# Achieving Structurally Robust Gromov-Wasserstein Distance via Adaptive Dual-Mask

Kangke Cheng [* 1]  Jiawei Huang [* 1 2]  Jingni Song [1]  Wanlin Zhang [1 3]  Bangxian Han [4]  Hu Ding [1]

## Abstract

The Gromov-Wasserstein (GW) distance enables comparison across different spaces but remains fragile to structural noise due to its global quadratic coupling. Existing robust extensions primarily rely on node-centric mass relaxation. However, we argue that this strategy is far from sufficient: it only addresses node-induced structural noise (outliers) while neglecting edge-induced distortions where spurious connections exist between valid nodes. To overcome this limitation, we propose the Structurally Robust Gromov-Wasserstein (SRGW) distance, a novel formulation that adaptively filters geometric distortions during optimization. By introducing a structure-aware dual-mask mechanism, our method effectively isolates these stubborn structural outliers while preserving strict marginal constraints for balanced transport. We solve this objective using a Mask-Guided GW Algorithm, which jointly optimizes the transport plan and the structural noise filters. We provide a rigorous theoretical analysis proving that our algorithm converges to a critical point under the Kurdyka–Łojasiewicz framework. Extensive experiments on synthetic geometric matching and real-world subgraph alignment benchmarks demonstrate that Mask-Guided GW achieves superior alignment quality, particularly under severe structural noise.

## 1. Introduction

Comparing complex, structured data objects is a fundamental challenge across machine learning, computer vision, and the natural sciences (Bronstein et al., 2017). Tasks such as aligning protein interaction networks and biological data in computational biology (Singh et al., 2008; Demetci et al., 2022), matching feature points or distributions in computer vision (Rubner et al., 2000; Sarlin et al., 2020), and analyzing semantic structures in natural language processing (Kusner et al., 2015; Zhao et al., 2019) all require the ability to quantify similarity in non-Euclidean spaces. The theory of Optimal Transport (OT) has been a powerful mathematical and computational framework for addressing these challenges (Peyré & Cuturi, 2019). Originating from Monge's formulation and later relaxed by Kantorovich's formulation (Kantorovich, 2006), OT seeks to find the most efficient transportation plan to move mass from one probability distribution to another. The modern development of OT has established profound connections between measure theory, Riemannian geometry, and optimization, providing a geometric toolkit for data comparison (Villani et al., 2008).

Classical OT requires measures to be on the same space. To overcome this and compare distributions across different spaces, the Gromov-Wasserstein (GW) distance (Mémoli, 2011a) aligns metric-measure spaces by preserving intra-domain geometry. Let $\mu$ and $\nu$ be probability distributions with intra-domain distance matrices $C$ and $D$, respectively. We define the set of valid couplings as $\Pi(\mu,\nu) = \{T \in \mathbb{R}_+^{n \times m} \mid T\mathbf{1}_m = \mu, T^\top\mathbf{1}_n = \nu\}$, where $\mathbf{1}_d$ denotes a $d$-dimensional vector of ones. Given a loss function $L$ (typically the squared loss $L(a,b) = (a-b)^2$), the GW distance is defined as:

$$\mathrm{GW}(\mu,\nu) = \min_{T \in \Pi(\mu,\nu)} \sum_{i,j,k,l} L(C_{ij}, D_{kl}) T_{ik} T_{jl}. \quad (1)$$

Relying solely on relational structures, GW is isometry-invariant and widely used for tasks like 3D shape matching (Mémoli, 2011a; Zhang et al., 2021), graph alignment (Li et al., 2023; Xu et al., 2019a; 2021), and cross-lingual alignment in NLP (Alvarez-Melis & Jaakkola, 2018).

Despite its theoretical elegance, standard GW is extremely sensitive to noise and outliers. While recent works have addressed computational costs (Loiola et al., 2007; Li et al., 2023; Scetbon et al., 2022; Titouan et al., 2019b; Solomon et al., 2016), robustness remains a critical issue. The vulnerability stems from the hard marginal constraints,

---

[*]Equal contribution [1]School of Computer Science and Technology, University of Science and Technology of China, Hefei, Anhui, China [2]Department of Computer Science, City University of Hong Kong [3]Shanghai Innovation Institute, Shanghai, China [4]Shandong University, Jinan, Shandong, China. Correspondence to: Hu Ding <huding@ustc.edu.cn>.

*Proceedings of the 43$^{rd}$ International Conference on Machine Learning*, Seoul, South Korea. PMLR 306, 2026. Copyright 2026 by the author(s).

$T \in \Pi(\mu, \nu)$, which rigidly enforce full mass conservation. This compels the transport plan to map the entire source geometry to the target, even in the presence of structural noise or outliers. Due to the quadratic nature of the GW objective (1), this forced alignment is particularly damaging: a single corrupted relationship $C_{ij}$ acts as a flawed reference frame, propagating errors globally rather than locally.

We demonstrate this fragility in Figure 1: by plotting the node-wise structural stress (contribution to the total loss), we observe that the error is not localized to the outlier. Instead, due to the non-local nature of the quadratic coupling, the stress propagates globally across the source graph (shifting from blue to green/yellow), effectively inflating the distance and degrading the matching fidelity for inliers.

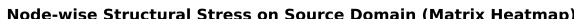

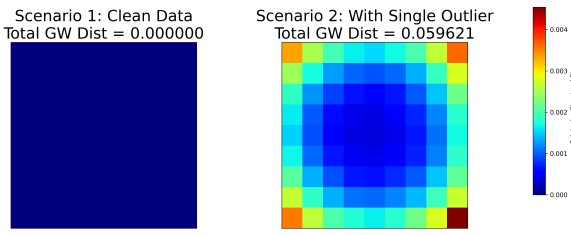

*(a)* Global Error Propagation

*Figure 1.* **Illustration of GW Sensitivity to Noise.** The quadratic objective causes this error to propagate globally, increasing structural stress across the entire graph instead of remaining localized.

## Our Contributions

To mitigate this issue, the literature has introduced relaxed variants of the Gromov–Wasserstein problem, most notably the Unbalanced Gromov-Wasserstein (UGW) (Séjourné et al., 2021) and Partial Gromov-Wasserstein (PGW) (Chapel et al., 2020b) frameworks. These methods achieve robustness by modifying the feasible set $\Pi(\mu, \nu)$. Instead of enforcing exact marginal constraints $T\mathbf{1} = \mu$, UGW relaxes them via soft penalties (e.g., the Kullback–Leibler divergence $D_{KL}(T\mathbf{1} \mid \mu)$), while PGW imposes a hard constraint on the total transported mass to be strictly less than 1. Kong et al. (2023) also propose Robust Gromov-Wasserstein (RGW), a formulation that introduces perturbed marginal constraints based on Kullback-Leibler divergence. However, these methods largely inherit their robust mechanisms directly from linear Optimal Transport (OT) theory like Partial OT (Chapel et al., 2020a) and Unbalanced OT (Fatras et al., 2021).

While effective in linear settings, we argue that transplanting marginal relaxation mechanisms to GW ignores a fundamental dichotomy: Linear OT aligns features, whereas GW aligns structures.

In Linear OT, marginal relaxation effectively isolates out-

liers due to the independence of matching costs. However, GW faces structural noise—defined as any deviation from isometry. This encompasses not only explicit outlier nodes but, more critically, corrupted geometric relationships (e.g., spurious edges) between valid inliers, which node-centric relaxation fails to isolate.

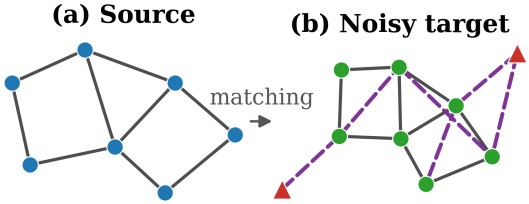

*Figure 2.* The left panel shows a clean source graph, while the right panel shows a noisy target graph containing valid nodes, outlier nodes, preserved edges, and spurious edges. Although node-centric marginal relaxation can suppress outlier nodes, it cannot selectively remove spurious edges between valid nodes without discarding useful structural information. This motivates our edge-level masking mechanism for filtering noisy structural interactions during matching.

To overcome the limitations of node-centric robustness, we adopt an edge-to-edge matching perspective. Unlike linear Optimal Transport, which matches nodes independently, Gromov-Wasserstein aligns the source edge $(i, j)$ to the target edge $(k, l)$ via the quadratic coupling term $T_{ik}T_{jl}$. This interdependence allows structural noise—such as large discrepancies in $|C_{ij} - D_{kl}|^2$—to propagate globally. Consequently, even if an outlier node is eventually discarded, its corrupted edges generate erroneous gradients that distort the alignment of valid inliers. To address this, we propose edge-centric interaction decoupling. Instead of relaxing marginal constraints, we maintain $T \in \Pi(\mu, \nu)$ to preserve global geometry, while introducing auxiliary variables to explicitly identify and filter structural noise at the edge level. Realizing this involves two key challenges:

- ***Challenge I: Efficient Optimization.*** The resulting non-convex objective implies potential computational bottlenecks. We require a solver that achieves robustness without sacrificing the speed expected of modern OT applications.

- ***Challenge II: Theoretical Convergence.*** Given the coupled nature of the objective, ensuring mathematical soundness is difficult. We must strictly prove that our iterative solver converges to a stationary point.

To address Challenge I, we reveal that the filtering sub-problem reduces to a Continuous Knapsack Problem, en-

abling an efficient closed-form solution via our Mask-Guided GW algorithm. Addressing Challenge II, we cast the joint update of the transport plan and edge masks as an inexact block coordinate scheme and analyze it within the Kurdyka–Łojasiewicz (KL) framework for nonconvex optimization (Attouch et al., 2013). By establishing a sufficient-decrease property and a relative subgradient error bound that tolerate inexact transport solves, we prove that the iterates have finite length and converge to a critical point of the objective; furthermore, leveraging the KL exponent at the limit point, we derive explicit local convergence rates, ranging from linear to sublinear depending on the local geometry.

## 2. Our Formulation: Structurally Robust Gromov-Wasserstein Distance

To overcome the barrier, we propose a paradigm shift from node-wise mass disposal to edge-wise interaction filtering. Building directly upon the standard GW formulation in (1), our key innovation is to decouple the transport decision from the reliability evaluation, leading to the **Structurally Robust Gromov-Wasserstein (SRGW)** Distance.

**Variables and Constraints Definitions.** Before presenting the objective, we explicitly define the auxiliary mechanism used to filter structural noise:

- **Noise Filters** $(\alpha, \beta)$**:** We introduce two sparse filtering matrices $\alpha \in \mathbb{R}^{n \times m}$ and $\beta \in \mathbb{R}^{n \times m}$. These variables act as adaptive masks that selectively absorb the error from high-distortion interactions, with entrywise constraints $0 \leq \alpha, \beta \leq T$ so each mask removes no more mass than the corresponding transport entry.

- **Total Noise Budget** $(\epsilon_1, \epsilon_2)$**:** To prevent the model from discarding too much structural information (which would lead to trivial solutions), we constrain the total magnitude of these filters using thresholds $\epsilon_1, \epsilon_2 \geq 0$.

- **Sparsity Constraint** $(\|\cdot\|_1)$**:** We employ the entry-wise $L_1$ norm, denoted as $\|\cdot\|_1$ (where $\|\alpha\|_1 = \sum_{i,j} |\alpha_{ij}|$), to measure the capacity of the filters. This constraint forces the filters to be *sparse*, ensuring that they only target a small subset of corrupted edges (outliers) while leaving the majority of the clean structure strictly preserved.

By incorporating these sparse filters directly into the quadratic coupling term, we arrive at the SRGW formulation:

$$\min_{\substack{T \in \Pi(\mu, \nu) \\ 0 \leq \alpha \leq T, \ 0 \leq \beta \leq T \\ \|\alpha\|_1 \leq \epsilon_1, \ \|\beta\|_1 \leq \epsilon_2}} \mathrm{SRGW}(T, \alpha, \beta)$$

$$:= \sum_{i,j,k,l} L(C_{ij}, D_{kl})(T_{ik} - \alpha_{ik})(T_{jl} - \beta_{jl}). \quad (2)$$

Here, strictly enforcing $T \in \Pi(\mu, \nu)$ guarantees global mass conservation, while the terms $\alpha$ and $\beta$ perform the surgical attenuation of noisy edges under the global budget $\epsilon$. The capacity constraints $0 \leq \alpha, \beta \leq T$ are entrywise, so the effective masses $(T - \alpha)$ and $(T - \beta)$ remain nonnegative throughout optimization. These terms effectively allow the solver to "blind" itself to specific, high-error quadratic interactions. Consequently, the contribution of this noisy edge to the total objective is neutralized, preventing it from dominating the gradient and distorting the optimization landscape. This dual mechanism enables the optimization to focus exclusively on aligning the underlying clean structure, absorbing the error propagated by outliers without compromising the integrity or convergence of the primary transport plan $T$.

The masking mechanism is an interaction-level filter in factorized form. Rather than introducing an explicit 4-way reliability tensor over all edge pairs $((i, j), (k, l))$, which would require $O(n^2 m^2)$ additional variables, SRGW modulates the two coupling factors already present in the GW quadratic term through $(T_{ik} - \alpha_{ik})(T_{jl} - \beta_{jl})$. This keeps the filtering operation inside the structural interaction term while preserving a tractable $n \times m$ mask representation.

This formulation offers a structurally profound advantage over standard relaxations:

- **Strict Structural Integrity** $(T \in \Pi)$: We ensure that the global geometric structure is fully preserved.

- **Surgical Attenuation** $(\alpha, \beta)$: Instead of discarding data points, these variables selectively attenuate the contribution of corrupted interactions to the total cost.

## 3. The Algorithm: Mask-Guided GW

The design of our optimization algorithm directly reflects the proposed edge-centric decoupling paradigm. By assigning distinct roles to the transport plan $T$ and the masking variables $(\alpha, \beta)$, we organize the optimization into two alternating update blocks corresponding to *structural alignment* and *interaction filtering*. We adopt a block coordinate descent (BCD) framework (Tseng, 2001), termed Mask-Guided GW and summarized in Algorithm 1, to iteratively address the intrinsic coupling between these components.

- **Phase 1 (Structural Alignment):** Fix the masks to

filter out currently identified noise, and update $T$ to find the best geometric overlay.

- **Phase 2 (Interactive Filtering):** Fix the transport $T$, and update $\alpha, \beta$ to identify high-distortion edges based on the current alignment.

---

**Algorithm 1** Mask-Guided GW

---

1: **Input:** Distance matrices $C \in \mathbb{R}^{n \times n}, D \in \mathbb{R}^{m \times m}$; marginals $\mu \in \Delta_n, \nu \in \Delta_m$; outlier budgets $\epsilon_1, \epsilon_2$.
2: **Initialize:** $T^{(0)} \leftarrow \mu\nu^T$, $\alpha^{(0)} \leftarrow \mathbf{0}$, $\beta^{(0)} \leftarrow \mathbf{0}$.
3: **for** $k = 0, 1, 2, \ldots$ until convergence **do**
4:    **T-step:** Update the transport plan by approximately solving the $T$-subproblem until a prescribed accuracy $\varepsilon_k > 0$ is reached
5:    $\alpha$-**step:** Compute the gain matrix $G^\alpha$ as in (6) (with $T = T^{(k+1)}$, $\beta = \beta^{(k)}$), then update $\alpha^{(k+1)} \leftarrow \text{GREEDYMASKSOLVER}(G^\alpha, T^{(k+1)}, \epsilon_1)$ (Algorithm 2).
6:    $\beta$-**step:** Compute $G^\beta$ symmetrically (with $T = T^{(k+1)}$, $\alpha = \alpha^{(k+1)}$), then update $\beta^{(k+1)} \leftarrow \text{GREEDYMASKSOLVER}(G^\beta, T^{(k+1)}, \epsilon_2)$.
7: **end for**
8: **Return:** $T^{(k+1)}, \alpha^{(k+1)}, \beta^{(k+1)}$.

---

**Algorithm 2** Greedy Mask Solver

---

1: **Input:** Gain matrix $G \in \mathbb{R}^{n \times m}$, capacity matrix $T \in \mathbb{R}_+^{n \times m}$, budget $\epsilon \geq 0$.
2: Sort the index pairs $\{(i, k)\}$ in descending order of $G_{ik}$, and denote the sorted sequence by $(i_1, k_1), (i_2, k_2), \ldots, (i_{nm}, k_{nm})$.
3: Initialize $M \leftarrow \mathbf{0}_{n \times m}$ and residual budget $r \leftarrow \epsilon$.
4: **for** $s = 1, 2, \ldots, nm$ **do**
5:    **if** $r \leq 0$ **or** $G_{i_s k_s} \leq 0$ **then**
6:       **break**
7:    **end if**
8:    $M_{i_s k_s} \leftarrow \min(T_{i_s k_s}, r)$.
9:    $r \leftarrow r - M_{i_s k_s}$.
10: **end for**
11: **Output:** Mask matrix $M$.

---

### 3.1. T-step: Inexact Update via Modified BAPG

In the T-step, we fix $\alpha = \alpha^{(k)}$ and $\beta = \beta^{(k)}$ and solve for $T$. The subproblem is:

$$
\min_{T \in \Pi(\mu,\nu)} \quad \mathcal{L}_T(T) \\
:= \sum_{i,j,k,l} L(C_{ij}, D_{kl})(T_{ik} - \alpha_{ik})(T_{jl} - \beta_{jl}). \quad (3)
$$

This is a quadratic program with a dense quadratic term and a linear term. To solve it efficiently, we run a Bregman

Alternating Projected Gradient (BAPG) procedure (Li et al., 2023) until a prescribed accuracy level $\varepsilon_k$ is met (see Assumption C.3 for the inexactness conditions). In practice, we terminate the inner BAPG routine once the stationarity residual falls below $\varepsilon_k$: $\text{dist}\big(0, \nabla_T \mathcal{L}(T, \alpha^{(k)}, \beta^{(k)}) + \mathcal{N}_{\Pi(\mu,\nu)}(T)\big) \leq \varepsilon_k$. Here, $\mathcal{N}_{\Pi(\mu,\nu)}(T)$ denotes the normal cone to the feasible set $\Pi(\mu, \nu)$ at $T$, and the term $\nabla_T \mathcal{L} + \mathcal{N}_{\Pi(\mu,\nu)}$ represents the first-order optimality condition. A zero distance implies $T$ is a stationary point.

Crucially, our reliance on this residual-based termination is justified by the Luo-Tseng error bound property (Luo & Tseng, 1992) associated with the Gromov-Wasserstein objective (Li et al., 2023). As detailed in Lemma B.1 (in the Appendix B), this error bound guarantees that the distance from the current iterate $T$ to the set of stationary points is upper-bounded by the stationarity residual. Therefore, driving the residual below $\varepsilon_k$ effectively ensures the iterate is within an $O(\varepsilon_k)$-neighborhood of the optimal solution set, satisfying the inexactness requirement (Assumption C.3) of our outer loop.

**Bilinearization via Constraint Decoupling.** To tackle the non-convex quadratic objective, we employ a bilinearization strategy by introducing an auxiliary variable $W$ subject to the consensus constraint $T = W$. This allows us to decouple the difficult joint constraints on the transport plan. Specifically, we reformulate the quadratic term $T_{ik}T_{jl}$ as the bilinear form $T_{ik}W_{jl}$, where we relax the constraints such that $T$ only satisfies the row marginals and $W$ only satisfies the column marginals:

$$
\mathcal{C}_1 := \{T \geq 0 : T\mathbf{1}_m = \mu\}, \\
\mathcal{C}_2 := \{W \geq 0 : W^\top \mathbf{1}_n = \nu\}, \quad (4)
$$

so that $T \in \mathcal{C}_1$ and $W \in \mathcal{C}_2$. This decomposition breaks the dependency between the two variables, enabling us to update each efficiently via closed-form solutions in an alternating minimization scheme.

**Penalty Method**: The bilinearization and the equality constraint $T = W$ are incorporated into a single penalized objective function. The constraint is enforced using a Bregman divergence term $\lambda D_h(T, W)$, where $\lambda > 0$ is a penalty parameter. The resulting subproblem at outer iteration $k$ (for fixed $\alpha = \alpha^{(k)}$ and $\beta = \beta^{(k)}$) becomes a joint minimization over $T$ and $W$:

$$
\min_{T,W \geq 0} \quad \mathcal{L}_\lambda(T, W) := \mathcal{L}_{\text{GW}}(T, W) + \lambda D_h(T, W).
$$

Here, $\mathcal{L}_{\text{GW}}(T, W)$ is the bilinearized objective derived from (3),

$$
\mathcal{L}_{\text{GW}}(T, W) = \sum_{i,j,k,l} L(C_{ij}, D_{kl})(T_{ik} - \alpha_{ik})(W_{jl} - \beta_{jl}),
$$

and $D_h(T, W)$ is the generalized Kullback-Leibler (KL) divergence,

$$D_h(T, W) = \sum_{i,k} \left( T_{ik} \log \frac{T_{ik}}{W_{ik}} - T_{ik} + W_{ik} \right).$$

This joint problem is then solved inexactly via the alternating updates described next.

**Alternating Updates**: We perform multiplicative updates for $T$ and $W$. Given the current estimates $T^{\text{old}}$ and $W^{\text{old}}$ from the previous inner or outer step, the new estimates are computed as:

$$T^{\text{new}} = \text{Proj}_{\mathcal{C}_1} \left[ T^{\text{old}} \odot \exp \left( -\frac{1}{\lambda} \nabla_T \mathcal{L}_{\text{GW}} \right) \right],$$

$$W^{\text{new}} = \text{Proj}_{\mathcal{C}_2} \left[ W^{\text{old}} \odot \exp \left( -\frac{1}{\lambda} \nabla_W \mathcal{L}_{\text{GW}} \right) \right],$$

where $\text{Proj}_{\mathcal{C}_1}$ and $\text{Proj}_{\mathcal{C}_2}$ are the Bregman projections onto the row- and column-marginal sets defined in (4). They admit a closed-form row/column rescaling:

$$\text{Proj}_{\mathcal{C}_1}(A) = \text{diag}\big(\mu \oslash (A\mathbf{1}_m)\big) A,$$
$$\text{Proj}_{\mathcal{C}_2}(B) = B \, \text{diag}\big(\nu \oslash (B^\top \mathbf{1}_n)\big).$$

The gradients are derived from the bilinearized objective:

$$(\nabla_T \mathcal{L}_{\text{GW}})_{ik} = \sum_{j,l} L(C_{ij}, D_{kl})(W_{jl}^{\text{old}} - \beta_{jl}),$$

$$(\nabla_W \mathcal{L}_{\text{GW}})_{jl} = \sum_{i,k} L(C_{ij}, D_{kl})(T_{ik}^{\text{new}} - \alpha_{ik}).$$

The output of this step, $T^{\text{new}}$, becomes $T^{(k+1)}$ for the outer loop.

### 3.2. $\alpha$-step and $\beta$-step: Greedy Solutions

For the $\alpha$-step, we fix $T = T^{(k+1)}$ and $\beta = \beta^{(k)}$. The subproblem for $\alpha$ is:

$$\max_\alpha \quad \sum_{i,k} \alpha_{ik} \Big( \sum_{j,l} L(C_{ij}, D_{kl})(T_{jl}^{(k+1)} - \beta_{jl}^{(k)}) \Big)$$

$$\text{s.t.} \quad 0 \le \alpha_{ik} \le T_{ik}^{(k+1)} \quad \forall i, k, \quad \text{and} \quad \sum_{i,k} \alpha_{ik} \le \epsilon_1.$$

**Optimization of Noise Filters.** Since the objective function is bilinear, fixing the transport plan $T$ transforms the $\alpha$-subproblem into a Continuous Knapsack Problem. We aim to maximize the linear profit defined by the gain matrix, subject to element-wise box constraints and a global budget:

$$\max_{0 \le \alpha_{ik} \le T_{ik}^{(k+1)}} \sum_{i,k} \alpha_{ik} G_{ik}^\alpha, \quad \text{s.t.} \sum_{i,k} \alpha_{ik} \le \epsilon_1, \quad (5)$$

where the gain matrix $G^\alpha$ represents the gradient of the loss, quantifying the potential cost reduction for each interaction:

$$G_{ik}^\alpha = \sum_{j,l} L(C_{ij}, D_{kl})(T_{jl}^{(k+1)} - \beta_{jl}^{(k)}). \quad (6)$$

This problem admits an exact and efficient solution via a greedy strategy (Algorithm 2, presented above alongside Algorithm 1). We sort the entries of $G^\alpha$ in descending order and sequentially allocate the budget $\epsilon_1$ to the entries with the highest gains, saturating each $\alpha_{ik}$ up to its upper bound $T_{ik}^{(k+1)}$ until the budget is exhausted. The $\beta$-step is strictly symmetric: we compute the corresponding gain $G^\beta$ and apply the same greedy allocation under budget $\epsilon_2$.

## 4. Theoretical Analysis

We establish convergence guarantees for Mask-Guided GW under the Kurdyka–Łojasiewicz (KL) framework for non-convex block coordinate methods (Attouch et al., 2013). The analysis proceeds in three steps. Section 4.1 introduces the analytic objects used throughout: an extended objective $\Phi$ that absorbs the feasibility constraints into a single nonsmooth function, and a proximalized variant of the $\alpha$- and $\beta$-steps that satisfies the standard "sufficient decrease + relative error" conditions required by the KL framework. Section 4.2 states our main theorem on the iterates of this proximal variant, covering both whole-sequence convergence to a critical point of $\Phi$ and KL-based local rates. Section 4.3 establishes the theorem via two lemmas (sufficient decrease and a relative subgradient bound) on the objects defined in Section 4.1; detailed proofs of the lemmas are deferred to the Appendix.

### 4.1. Setup: Extended Objective and Proximal Mask Updates

**Extended objective.** Let $x := (T, \alpha, \beta)$ and recall the SRGW loss and feasible set from (2):

$$\mathcal{L}(x) := \sum_{i,j,k,l} L(C_{ij}, D_{kl}) (T_{ik} - \alpha_{ik}) (T_{jl} - \beta_{jl}),$$

$$\mathcal{D} := \big\{ x : T \in \Pi(\mu, \nu),\ 0 \le \alpha \le T,\ 0 \le \beta \le T,$$
$$\langle \mathbf{1}, \alpha \rangle \le \epsilon_1,\ \langle \mathbf{1}, \beta \rangle \le \epsilon_2 \big\},$$

where inequalities are entrywise and $\langle \mathbf{1}, \alpha \rangle := \sum_{i,k} \alpha_{ik}$. To treat $\mathcal{L}$ and the constraints jointly, we work with the extended objective

$$\Phi(x) := \mathcal{L}(x) + \delta_{\mathcal{D}}(x),$$

where $\delta_{\mathcal{D}}$ is the indicator function of $\mathcal{D}$. The KL framework tracks the descent of $\Phi$ along $\{x^{(k)}\}$ and certifies convergence through subdifferential bounds on $\Phi$; both are invoked in the proof in Section 4.3.

**Proximalized mask updates.** The $T$-step in Algorithm 1 runs a fixed finite number of BAPG sweeps and is treated as an inexact oracle satisfying standard descent and relative-stationarity conditions (Assumption C.3). For the mask blocks, the KL analysis requires a proximalized variant of the greedy step. With the gain matrix $G^\alpha$ from (6) and a fixed proximal parameter $\delta > 0$, the $\alpha$-update reads

$$\alpha^{(k+1)} \in \arg \max_{\substack{0 \leq \alpha \leq T^{(k+1)} \\ \langle \mathbf{1}, \alpha \rangle \leq \epsilon_1}} \langle G^\alpha, \alpha \rangle - \frac{\delta}{2} \left\| \alpha - \alpha^{(k)} \right\|_F^2, \quad (7)$$

and analogously for $\beta$ with gain $G^\beta$:

$$\beta^{(k+1)} \in \arg \max_{\substack{0 \leq \beta \leq T^{(k+1)} \\ \langle \mathbf{1}, \beta \rangle \leq \epsilon_2}} \langle G^\beta, \beta \rangle - \frac{\delta}{2} \left\| \beta - \beta^{(k)} \right\|_F^2. \quad (8)$$

Equivalently, (7) is the Euclidean projection $\alpha^{(k+1)} = \Pi_{\mathcal{A}(T^{(k+1)})}\big(\alpha^{(k)} + \frac{1}{\delta} G^\alpha\big)$ onto $\mathcal{A}(T) := \{\alpha : 0 \leq \alpha \leq T, \langle \mathbf{1}, \alpha \rangle \leq \epsilon_1\}$, and analogously for $\beta$. As $\delta \downarrow 0$, these proximal steps recover the greedy solution computed by Algorithm 2 (used in practice for efficiency); the convergence guarantees in Section 4.2 require a fixed $\delta > 0$.

### 4.2. Main Convergence Theorem

**Theorem 4.1** (Convergence and rate of Mask-Guided GW). *Let $\{x^{(k)}\}_{k \geq 0}$ be the sequence generated by Mask-Guided GW with proximal mask updates under Assumptions C.1–C.3, where $x^{(k)} := \big(T^{(k)}, \alpha^{(k)}, \beta^{(k)}\big)$. We equip the product space of $(T, \alpha, \beta)$ with the Frobenius norm $\|\cdot\|_F$ by concatenation. Then the following hold:*

1. *__Finite length and convergence.__ The sequence has finite length, $\sum_{k=0}^\infty \left\| x^{(k+1)} - x^{(k)} \right\|_F < \infty$, and therefore $x^{(k)} \to x^\star$ for some critical point $x^\star$ of $\Phi$, i.e., $0 \in \partial \Phi(x^\star)$.*

2. *__Local rate via KL exponent.__ If $\Phi$ has KL exponent $\theta \in [0, 1)$ at $x^\star$, then the local rate is:*

   - *If $\theta \in (1/2, 1)$, $\|x^{(k)} - x^\star\|_F = \mathcal{O}\Big(k^{-\frac{1-\theta}{2\theta-1}}\Big)$.*
   - *If $\theta \in [0, 1/2]$, $\|x^{(k)} - x^\star\|_F \leq Cq^k$ for some $C > 0$ and $q \in [0, 1)$.*

*Remark* 4.2 (KL exponent). The KL exponent $\theta$ is a local geometric quantity of $\Phi$ at $x^\star$. Concretely, $\Phi$ is said to have KL exponent $\theta \in [0, 1)$ at $x^\star$ if there exist $c > 0$ and a neighborhood $\mathcal{U}$ of $x^\star$ such that for all $x \in \mathcal{U}$ with $\Phi(x^\star) < \Phi(x) < \Phi(x^\star) + \eta$, it holds that

$$\mathrm{dist}\big(0, \partial\Phi(x)\big) \geq c\big(\Phi(x) - \Phi(x^\star)\big)^\theta. \quad (9)$$

Smaller values of $\theta$ correspond to a sharper local landscape and typically yield faster (linear) convergence, whereas $\theta$ closer to 1 indicates a flatter geometry and results in sublinear rates.

Theorem 4.1 certifies whole-sequence convergence to a critical point of $\Phi$ together with KL-based local rates, despite (i) the nonconvex quadratic coupling, (ii) the cross-block constraint $0 \leq \alpha, \beta \leq T$, and (iii) the inexact $T$-update. Prior OT analyses typically rely on convexity, entropic regularization, or static marginals, none of which apply here. We instead develop a tailored inexact BCD/KL analysis built directly on the extended objective $\Phi$ and the proximal mask updates (7)–(8) from Section 4.1. The argument rests on two ingredients, both established in Section 4.3: a sufficient-decrease inequality for $\Phi$ along $\{x^{(k)}\}$ that absorbs the inner-solver inexactness (Lemma 4.3), and a relative subgradient bound that controls $\mathrm{dist}(0, \partial\Phi(x^{(k+1)}))$ by $\|x^{(k+1)} - x^{(k)}\|_F$ (Lemma 4.4). Combined with the KL property of $\Phi$, these yield finite length and the stated rates.

### 4.3. Proof of Theorem 4.1

Due to space limitations, all detailed proofs of the following lemmas are deferred to Appendix C.

**Lemma 4.3** (Sufficient decrease). *Let $\{x^{(k)}\}$ be the sequence produced by Mask-Guided GW with proximal mask steps (7)–(8). Under Assumptions C.1–C.3, there exists a constant $\rho_1 > 0$ such that for all $k \geq 0$,*

$$\Phi(x^{(k)}) - \Phi(x^{(k+1)}) \geq \rho_1 \left\| x^{(k+1)} - x^{(k)} \right\|_F^2 - \varepsilon_k.$$

**Lemma 4.4** (Subgradient bound). *Under the same conditions as Lemma 4.3, there exists a constant $\rho_2 > 0$ such that for all $k \geq 0$,*

$$\mathrm{dist}\big(0, \partial\Phi(x^{(k+1)})\big) \leq \rho_2 \left\| x^{(k+1)} - x^{(k)} \right\|_F + \varepsilon_k.$$

*Remark* 4.5. The convergence guarantees in Section 4 are stated for the proximalized mask updates (7)–(8) with a fixed $\delta > 0$. The greedy knapsack update can be viewed as the limiting case $\delta \downarrow 0$, and is used as an efficient heuristic in practice.

The proof relies on the convergence analysis for inexact descent methods satisfying the Kurdyka–Łojasiewicz property, specifically Theorem 2.9 in (Attouch et al., 2013). We proceed in three steps to verify the preconditions of this theorem.

**(1). Sufficient decrease and convergence of objective values.** Lemma 4.3 establishes the descent inequality

$$\Phi(x^{(k)}) - \Phi(x^{(k+1)}) \geq \rho_1 \|x^{(k+1)} - x^{(k)}\|_F^2 - E_k. \quad (10)$$

Since $\Phi$ is continuous and the constraint set $\mathcal{D}$ is compact (Assumption C.1), $\Phi$ is bounded from below by some value $\underline{\Phi} > -\infty$. Furthermore, the error sequence $\{E_k\}$ is summable. Summing the descent inequality from $k = 0$ to

$K$ yields

$$\sum_{k=0}^{K} \rho_1 \|x^{(k+1)} - x^{(k)}\|_F^2$$

$$\leq \Phi(x^{(0)}) - \Phi(x^{(K+1)}) + \sum_{k=0}^{K} E_k < \infty. \quad (11)$$

This implies that $\|x^{(k+1)} - x^{(k)}\|_F \to 0$ as $k \to \infty$. Moreover, since $\Phi$ is bounded below and $\{E_k\}$ is summable, the sequence $\{\Phi(x^{(k)})\}$ converges to a finite limit value $\Phi^\star$.

**(2). Gradient bound and critical point.** Lemma 4.4 provides the relative error bound on the subgradient:

$$\text{dist}(0, \partial\Phi(x^{(k+1)})) \leq \rho_2 \|x^{(k+1)} - x^{(k)}\|_F + \varepsilon_k, \quad (12)$$

where $\{\varepsilon_k\}$ is summable. Let $\omega(x^{(0)})$ be the set of limit points of the sequence $\{x^{(k)}\}$. Since the sequence lies in a compact set $\mathcal{D}$, $\omega(x^{(0)})$ is nonempty. Let $x^\star \in \omega(x^{(0)})$ be a limit point such that a subsequence $x^{(k_j)} \to x^\star$. Since $\|x^{(k+1)} - x^{(k)}\|_F \to 0$ and $\varepsilon_k \to 0$, the subgradient distance vanishes: $\text{dist}(0, \partial\Phi(x^{(k_j)})) \to 0$. Due to the closedness of the graph of the limiting subdifferential $\partial\Phi$ and the continuity of $\Phi$, we conclude that $0 \in \partial\Phi(x^\star)$, i.e., $x^\star$ is a critical point.

**(3). Finite length and convergence rates.** By Assumption C.1, $\Phi$ is a semi-algebraic function, which ensures it satisfies the KL property at $x^\star$ with some exponent $\theta \in [0, 1)$. Combining the sufficient decrease (with summable error $E_k$) and the subgradient bound (with summable error $\varepsilon_k$), the sequence $\{x^{(k)}\}$ satisfies all conditions of the inexact KL convergence theorem (Theorem 2.9 in Attouch et al., 2013). This theorem guarantees that the trajectory has finite length:

$$\sum_{k=0}^{\infty} \|x^{(k+1)} - x^{(k)}\|_F < \infty. \quad (13)$$

The finite length property implies that $\{x^{(k)}\}$ is a Cauchy sequence, and thus the entire sequence converges to the critical point $x^\star$. Finally, the local convergence rates follow directly from the algebraic form of the desingularizing function $\varphi(s) = cs^{1-\theta}$ associated with the KL exponent $\theta$. Specifically, if $\theta \in (1/2, 1)$, the rate is polynomial, and if $\theta \in [0, 1/2]$, the rate is linear, as stated in the theorem.

# 5. Experiments

In this section, we present comprehensive experimental results to validate the effectiveness of our proposed Mask-Guided GW algorithm on 2D geometric matching and graph alignment tasks. Due to space constraints, more exhaustive experimental setups, additional datasets, and further experimental results are provided in the Appendix C.1. All implementations were executed on a server running Ubuntu 22.04.1, powered by an Intel Xeon E5-2680 v4 CPU @ 2.40GHz and Tesla P100-PCIE-16GB*4.

**Baselines** We compare Mask-Guided GW against a comprehensive set of robust and unbalanced GW solvers:

- **Unbalanced/Robust Methods:** UGW (Unbalanced GW) (Séjourné et al., 2021), RGW (Outlier-Robust GW) (Kong et al., 2023), PGW (Partial GW) (Chapel et al., 2020b), MD-SRGW (Semi-Relaxed GW) (Vincent-Cuaz et al., 2022).

- **Balanced/Standard Methods:** FW (Frank-Wolfe) (Titouan et al., 2019a), BPG (Xu et al., 2019b), eBPG (Solomon et al., 2016), BAPG (Li et al., 2023), and SpecGWL(Chowdhury & Needham, 2020).

## 5.1. Synthetic 2D Geometric Matching

**Setup.** Source points are sampled from geometric primitives. Targets are generated via random rigid transformations, scaling, and Gaussian noise. Crucially, we introduce structured outliers (∼20%) arranged in coherent patterns (rather than uniform noise) to maximize structural ambiguity and test robustness against deceptive clusters.

**Results.** Figure 3 visualizes the alignment. Existing methods (UGW, PGW) struggle with these structured outliers (red crosses), yielding diffuse coupling matrices with significant off-diagonal noise. RGW improves sparsity but retains artifacts. In contrast, Mask-Guided GW consistently recovers sharp, diagonal structures, demonstrating its superior ability to filter structural outliers and reconstruct the underlying geometry.

## 5.2. Graph Alignment Task

### 5.2.1. SUBGRAPH ALIGNMENT TASK

**Datasets and Setup.** Following the experimental protocol in (Li et al., 2023). We evaluate on four benchmarks: **BZR** (Sutherland et al., 2003), **DHFR** (Sutherland et al., 2003), **ENZYMES** (Borgwardt et al., 2005), and **Fingerprint** (Riesen & Bunke, 2008). To simulate partial matching, we construct target subgraphs by randomly sampling 50% of nodes ($\rho = 0.5$) from the original graphs. Crucially, we employ attribute-based geometry: instead of graph topology, the intra-domain distance matrices $D$ are computed as the Euclidean distance between node attribute vectors. This challenges solvers to align structures defined in high-dimensional feature spaces.

**Implementation Details.**

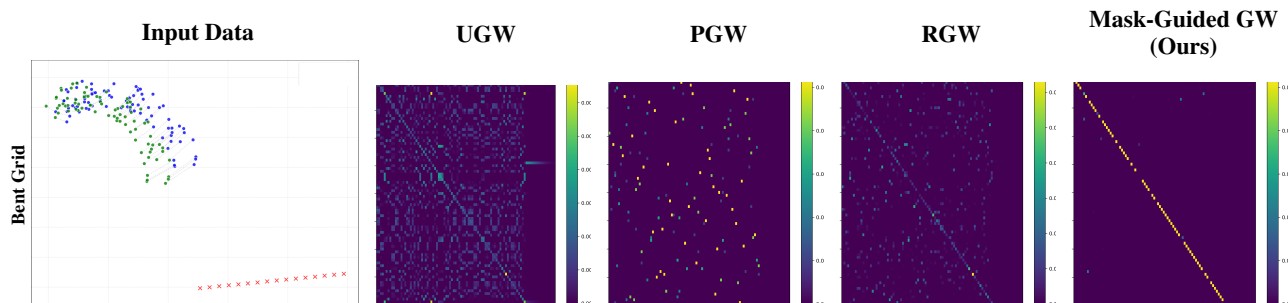

*Figure 3.* **Comparison of robustness across datasets.** The first column (**Input Data**, enlarged for visibility) visualizes the source (blue) and target (green/red) distributions. The subsequent columns display the coupling matrices from different methods. Our **Mask-Guided GW** (rightmost) consistently exhibits the cleanest diagonal structure.

*Table 1.* **Unified performance benchmark.** Comparison on **Subgraph Alignment** ($\rho = 0.5$, four datasets) and **Structural Noise** ($\eta = 0.1$, Fingerprint). We report node accuracy ($N_{acc}$, %), edge accuracy ($E_{acc}$, %, structural noise only), and runtime ($T$, seconds). Best accuracies are in **bold**.

| Method | Subgraph Alignment ($\rho = 0.5$) | | | | | | | | Structural Noise (Fingerprint) | | |
| | BZR | | Enzymes | | DHFR | | Fingerprint | | $\eta = 0.1$ | | |
| | $N_{acc}$ | $T$ | $N_{acc}$ | $T$ | $N_{acc}$ | $T$ | $N_{acc}$ | $T$ | $N_{acc}$ | $E_{acc}$ | $T$ |
|---|---|---|---|---|---|---|---|---|---|---|---|
| ***Balanced / Standard Methods*** | | | | | | | | | | | |
| FW | 14.0 | 0.8 | 7.2 | 0.5 | 10.2 | 1.3 | 15.7 | 1.3 | 15.7 | 3.7 | 1.5 |
| SpecGWL | 11.5 | 15.3 | 10.3 | 2.5 | 8.5 | 7.2 | 19.4 | 5.9 | 19.4 | 14.5 | 6.7 |
| eBPG | 13.8 | 11000 | 14.2 | 5900 | 8.3 | 12000 | 37.5 | 14000 | 37.3 | 31.3 | 16701 |
| BPG | 17.5 | 34.1 | 14.3 | 17.3 | 10.2 | 38.5 | 31.2 | 46.9 | 31.0 | 20.8 | 52.3 |
| BAPG | 18.8 | 15.7 | 30.6 | 8.6 | 12.1 | 24.0 | 40.2 | 8.8 | 40.1 | 37.3 | 9.7 |
| ***Unbalanced / Robust Methods*** | | | | | | | | | | | |
| MD-SRGW | 23.6 | 117 | 28.6 | 110 | 12.0 | 261 | 24.3 | 241 | 24.1 | 14.5 | 262 |
| UGW | 31.3 | 316 | 26.5 | 185 | 14.8 | 388 | 43.3 | 1904 | 43.3 | 47.3 | 2078 |
| PGW | 15.2 | 2.4 | 18.8 | 1.1 | 11.2 | 4.7 | 35.7 | 2.3 | 35.8 | 29.6 | 2.6 |
| RGW | 42.6 | 2652 | 41.6 | 1142 | 23.0 | 4945 | 25.9 | 1704 | 25.5 | 16.6 | 1981 |
| **Mask-Guided GW (Ours)** | **52.1** | 1599 | **52.3** | 77.1 | **34.1** | 265 | **47.5** | 234 | **50.9** | **60.8** | 392 |

- **Initialization:** We use uniform marginal distributions and initialize transport plans uniformly.

- **Hyperparameters:** For baselines (UGW, PGW, RGW, etc.), we perform extensive grid searches around the ranges recommended in their respective papers. For Mask-Guided GW, we fix $\epsilon_1 = \epsilon_2 = \epsilon$ and we perform a grid search for the structural noise budgets $\epsilon$ within the range $[0, 0.9]$.

- **Metric:** Performance is evaluated using standard matching accuracy (intersection over ground truth size).

**Main Results** The quantitative results across all four benchmarks (BZR, ENZYMES, DHFR, and Fingerprint) are summarized in Table 1.

#### 5.2.2. GRAPH EDGE ALIGNMENT UNDER STRUCTURAL NOISE

**Setup** . Unless otherwise specified, experimental settings follow the Subgraph Alignment Task. To simulate edge weight noise, we preserve the graph topology but perturb its geometry. Specifically, we select a fraction $\eta$ of existing edges and corrupt their weights (distances) with random noise, while maintaining node identities.

**Evaluation Metric: Edge Accuracy.** We measure topological preservation by the percentage of correctly matched edges. For each source edge $(i, j)$, the predicted match is the target edge maximizing the joint probability mass defined by the transport plan $P$. Edge Accuracy is the fraction of these predictions that align with the ground truth.

#### 5.2.3. ANALYSIS AND DISCUSSION

**Performance & Robustness.** As shown in Table 1, our method achieves state-of-the-art results across all benchmarks. For **Subgraph Alignment** ($\rho = 0.5$), we surpass

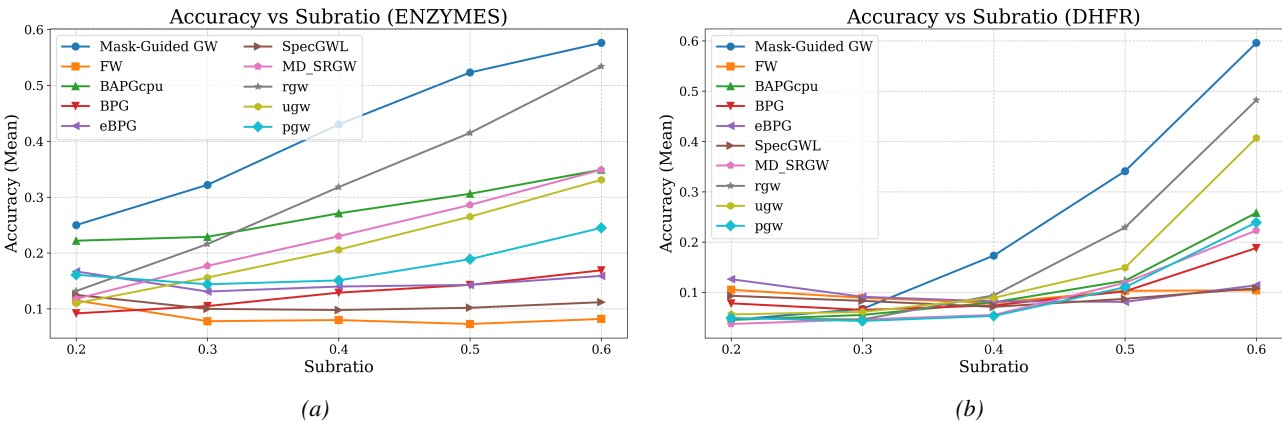

*Figure 4.* **Matching accuracy under varying subgraph ratios.** (a) Matching accuracy versus subgraph ratio on ENZYMES. (b) Matching accuracy versus subgraph ratio on DHFR.

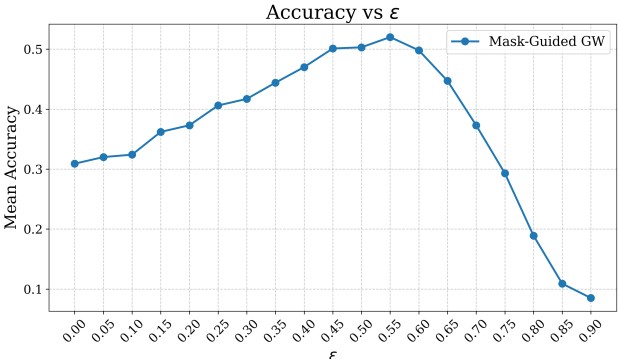

*Figure 5.* **Sensitivity analysis on ENZYMES with respect to $\epsilon$.**

the best baseline (RGW) by significant margins: $+10.7\%$ on Enzymes and $+11.1\%$ on DHFR. This confirms that our adaptive masking effectively handles heavy occlusion where mass-relaxation methods (UGW, PGW) struggle. Under **Structural Noise** (Fingerprint, $\eta = 0.1$), our method demonstrates superior topological resilience, achieving $60.8\%$ Edge Accuracy compared to UGW's $47.3\%$, proving its ability to filter spurious edge weights and preserve the geometric skeleton.

**Sensitivity & Efficiency.** Fig. 4 shows our method maintains the lead across varying ratios $\rho \in [0.2, 0.6]$, effectively leveraging increasing structural evidence while baselines like FW remain stagnant. The sensitivity curve (Fig. 5) peaks near the true inlier ratio ($\epsilon \approx 0.55$), acting as an effective prior. Computationally, our method offers the best trade-off: it is orders of magnitude faster than heavy solvers (e.g., $15\times$ faster than RGW on Enzymes) while delivering substantially higher accuracy than fast heuristics. Additional benchmark settings, parameter sensitivity, marginal feasibility, and theoretical properties are reported in Appendix C.3–C.5.

## 6. Conclusion

Gromov-Wasserstein (GW) matching is fragile precisely where it is most useful: its global quadratic coupling propagates a single corrupted pairwise relation across the entire transport plan. Existing robust variants inherit node-centric mass-relaxation from linear optimal transport, which can discard outlier nodes but cannot excise a spurious edge between two otherwise valid nodes. Our Structurally Robust GW (SRGW) instead reframes robustness at the interaction level: we keep strict couplings in $\Pi(\mu, \nu)$ to preserve global geometry and introduce two sparse, budgeted dual masks that filter individual edge interactions in factorized form. The resulting Mask-Guided GW alternates an inexact BAPG transport step with closed-form knapsack mask updates, and we prove finite-length convergence to a critical point under the Kurdyka–Łojasiewicz framework. Empirically, it attains state-of-the-art node and edge matching across synthetic and real benchmarks (Table 1), often by wide margins and at a fraction of the cost of heavier robust solvers. Our guarantees are local and the noise budgets $\epsilon_1, \epsilon_2$ are user-specified hyperparameters; future work includes scalable low-rank couplings, data-driven budget selection, and extending the edge-centric filter to fused, feature-aware GW.

## Impact Statement

This paper presents work whose goal is to advance the field of Machine Learning. There are many potential societal consequences of our work, none which we feel must be specifically highlighted here.

## Acknowledgements

The authors would like to thank the anonymous reviewers for their valuable comments and suggestions. This work was partially supported by the National Key Research and Development Program of China (No. 2021YFA1000900), the Na-

tional Natural Science Foundation of China (No. 62272432, No. 62432016), and the Natural Science Foundation of Anhui Province (No. 2208085MF163). The experiments and model training were supported by the robotic AI-Scientist platform of Chinese Academy of Sciences.

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

# A. Appendix.

## A.1. Related work

**Optimal Transport** The theory of Optimal Transport (OT) provides a powerful geometric framework for comparing probability distributions. Originating from the Monge problem and later relaxed by Kantorovich (Kantorovich, 2006), OT seeks a transport plan that minimizes the total cost of moving mass from a source to a target distribution. In the discrete setting, this corresponds to the Earth Mover's Distance (EMD) (Rubner et al., 2000), which has been widely applied in computer vision and machine learning. While historically plagued by cubic computational complexity, the field was revolutionized by the introduction of entropic regularization (Cuturi, 2013). This formulated the problem as strictly convex and enabled efficient solutions via the Sinkhorn-Knopp algorithm, allowing OT to scale to large-scale data applications (Peyré & Cuturi, 2019).

**Gromov-Wasserstein Distance and Efficient Computation** The Gromov-Wasserstein (GW) distance, introduced by Mémoli (Mémoli, 2011a), extends the classical Optimal Transport (OT) framework to compare probability measures defined on distinct metric spaces. Unlike the Wasserstein distance, which requires a ground cost across domains, GW minimizes the distortion of intra-domain pairwise distances, making it invariant to isometries such as rotation and translation. This property has established GW as a fundamental tool for structured data analysis, including 3D shape matching (Mémoli, 2011b), graph alignment (Xu et al., 2019b), and cross-lingual correspondence (Alvarez-Melis & Jaakkola, 2018).

Despite its theoretical elegance, the standard GW problem involves solving a non-convex Quadratic Assignment Problem (QAP), which is known to be NP-hard. To address the computational bottlenecks, significant efforts have been directed towards entropic regularization. Peyré et al. (2016) proposed an efficient Sinkhorn-based algorithm using heat kernels, which reduced the complexity of computing entropic GW. Subsequent works have further improved scalability through low-rank approximations (Scetbon et al., 2022) and sliced variants (Titouan et al., 2019b). While these methods successfully alleviate the computational burden, they largely adhere to the standard GW formulation, which assumes strict mass conservation and noise-free structural correspondence. Consequently, these efficient solvers remain brittle when applied to real-world scenarios characterized by outliers and background noise.

**Robustness in Optimal Transport** In the linear Optimal Transport setting, robustness against noise and outliers has been extensively studied. The strict mass conservation constraint in classical OT forces the transport plan to match outliers in the source domain to valid points in the target domain, yielding high transportation costs and erroneous mappings. To mitigate this, *Partial Optimal Transport* (Partial OT) (Chapel et al., 2020b) relaxes the equality constraints on marginals, allowing the transport plan to move only a fraction of the total mass (e.g., mass $s < 1$). Alternatively, *Unbalanced Optimal Transport* (UOT) (Fatras et al., 2021; Chizat et al., 2018) replaces the hard marginal constraints with soft penalties, such as the Kullback-Leibler (KL) divergence or Csiszár divergences. These formulations allow for the creation and destruction of mass, effectively handling mass variation and outliers. However, extending these "node-wise" relaxation mechanisms to the quadratic GW problem is non-trivial due to the complex interdependence of structural costs.

**Robustness in Gromov-Wasserstein Distance** Addressing the sensitivity of GW to structural noise is an active area of research. Early attempts focused on extending linear robust OT concepts to the quadratic setting. **Partial Gromov-Wasserstein (PGW)** (Chapel et al., 2020b) introduces a hard constraint on the total transported mass, optimizing for a sub-coupling that matches only the most similar substructures. While effective for partial matching, the hard mass constraint introduces a non-convexity that is difficult to tune and optimize. **Unbalanced Gromov-Wasserstein (UGW)** (Séjourné et al., 2021) formulates a relaxed objective by penalizing marginal deviations using KL divergence. This approach, often solved via a tensor product of unbalanced OT schemes, provides a mathematically sound framework for handling mass discrepancies. More recently, **Semi-Relaxed GW (MD-SRGW)** (Vincent-Cuaz et al., 2022) proposed relaxing only one marginal constraint to handle subgraph matching tasks where one structure is strictly contained in another. Additionally, Kong et al. (Kong et al., 2023) proposed an **Robust GW (RGW)** framework that explicitly models outlier sparse terms within the marginal constraints.

# B. Inner Loop Termination and Error Bound

In this section, we provide the theoretical justification for using the stationarity residual as the termination criterion for the inner BAPG solver. Specifically, we establish the relationship between the computable residual and the distance to the set of stationary points via the Luo-Tseng error bound condition.

### B.1. Stationarity Residual Definition

Consider the inner subproblem at the $k$-th outer iteration. For brevity, let $f(T)$ denote the objective function of the inner problem (which includes the quadratic Gromov-Wasserstein term and the linear terms from the Lagrangian multipliers). The feasible set is the transport polytope $\mathcal{C} = \Pi(\mu, \nu)$.

The first-order optimality condition for a point $T^* \in \mathcal{C}$ to be a stationary point is:

$$0 \in \nabla f(T^*) + \mathcal{N}_{\mathcal{C}}(T^*), \tag{14}$$

where $\mathcal{N}_{\mathcal{C}}(T)$ denotes the normal cone to the convex set $\mathcal{C}$ at $T$, defined as:

$$\mathcal{N}_{\mathcal{C}}(T) = \{G \in \mathbb{R}^{n \times m} \mid \langle G, Y - T \rangle \leq 0, \forall Y \in \mathcal{C}\}. \tag{15}$$

Based on proximal gradient theory, the condition in (14) is equivalent to the fixed-point equation involving the projection operator $\mathcal{P}_{\mathcal{C}}$:

$$T^* = \mathcal{P}_{\mathcal{C}}\left(T^* - \gamma \nabla f(T^*)\right), \quad \forall \gamma > 0. \tag{16}$$

Consequently, a natural measure of stationarity violation (the residual) at a current iterate $T$ is defined as the magnitude of the proximal gradient step:

$$\mathcal{R}(T) := \frac{1}{\gamma} \left\| T - \mathcal{P}_{\mathcal{C}}\left(T - \gamma \nabla f(T)\right) \right\|_F. \tag{17}$$

Note that $\mathcal{R}(T) = 0$ if and only if $T$ is a stationary point. In our implementation, we terminate the inner loop when $\mathcal{R}(T) \leq \varepsilon_k$.

### B.2. Luo-Tseng Error Bound Condition

To ensure the convergence of the outer inexact ADMM framework, we require that the iterate $T$ is sufficiently close to the set of true stationary points, denoted by $\mathcal{S}^*$. The following lemma bridges the gap between the computable residual $\mathcal{R}(T)$ and the distance to the solution set $\text{dist}(T, \mathcal{S}^*) := \inf_{T^* \in \mathcal{S}^*} \|T - T^*\|_F$.

**Lemma B.1** (Error Bound for GW). *Let $f(T)$ be the objective function of the Gromov-Wasserstein problem over the transport polytope $\mathcal{C}$. Since $f(T)$ is a quadratic function and $\mathcal{C}$ is a polyhedral set, the problem satisfies the Luo-Tseng error bound condition (Luo & Tseng, 1992).*

*Specifically, as shown in Li et al. (2023, Proposition 3.1), for any feasible $T \in \mathcal{C}$ within a neighborhood of the stationary set $\mathcal{S}^*$, there exist constants $\tau > 0$ and $\bar{\delta} > 0$ such that if $\mathcal{R}(T) \leq \bar{\delta}$, then:*

$$\text{dist}(T, \mathcal{S}^*) \leq \tau \cdot \mathcal{R}(T). \tag{18}$$

*Proof.* The proof follows directly from the structural properties of the problem. The objective function of the inner subproblem is a non-convex quadratic function (due to the GW term), and the constraints are linear equality and non-negativity constraints (defining a polytope).

According to Luo & Tseng (1992), coordinate descent or gradient projection methods applied to problems with this specific structure (quadratic objective over polyhedral constraints) admit a local error bound. This property guarantees that the distance to the set of stationary points is linearly bounded by the norm of the projected gradient residual.

Recently, Li et al. (2023) explicitly verified this condition for the specific case of Gromov-Wasserstein learning (refer to their Proposition 3.1 and Appendix C.1). By invoking this result, we conclude that driving the residual $\mathcal{R}(T)$ below $\varepsilon_k$ implies:

$$\text{dist}(T, \mathcal{S}^*) \leq \tau \varepsilon_k. \tag{19}$$

This ensures that the inexactness condition required for the outer loop convergence is satisfied up to a constant scaling factor. $\qquad\square$

## C. Missing Proofs

**Assumption C.1** (Regularity and KL property). Let $x = (T, \alpha, \beta)$ and $\Phi(x) := \mathcal{L}(T, \alpha, \beta) + \delta_{\mathcal{D}}(x)$. The feasible set $\mathcal{D}$ is nonempty, compact, and semi-algebraic. Moreover, $\mathcal{L}$ is a polynomial in $(T, \alpha, \beta)$. Consequently, $\Phi$ is semi-algebraic and satisfies the KL property at every point in $\mathrm{dom}(\Phi)$.

**Assumption C.2** (Strict positivity for Bregman geometry). The marginals lie in the relative interior of the simplices, i.e., $\min_i \mu_i > 0$ and $\min_k \nu_k > 0$. The inner $T$-solver is initialized with a strictly positive matrix $T^{(0)}$ and generates strictly positive iterates: there exists $c > 0$ such that for all outer iterations $k$ and all indices $(i, k)$,

$$T_{ik}^{(k)} \geq c. \tag{20}$$

**Assumption C.3** (Inexact $T$-updates with summable accuracy). Fix any $(\alpha, \beta)$ and define

$$\mathcal{L}_{\alpha,\beta}(T) := \mathcal{L}(T, \alpha, \beta) + \delta_{\Pi(\mu,\nu)}(T). \tag{21}$$

At outer iteration $k$, the $T$-step returns $T^{(k+1)} \in \Pi(\mu, \nu)$ satisfying, for a prescribed accuracy $\varepsilon_k > 0$,

$$\mathrm{dist}\Big(0, \ \nabla_T \mathcal{L}(T^{(k+1)}, \alpha^{(k)}, \beta^{(k)}) + \mathcal{N}_{\Pi(\mu,\nu)}(T^{(k+1)})\Big) \leq \varepsilon_k, \tag{22}$$

and there exists a constant $c_T > 0$, independent of $k$, such that

$$\mathcal{L}_{\alpha^{(k)},\beta^{(k)}}(T^{(k)}) - \mathcal{L}_{\alpha^{(k)},\beta^{(k)}}(T^{(k+1)}) \geq c_T \left\| T^{(k+1)} - T^{(k)} \right\|_F^2 - \varepsilon_k. \tag{23}$$

The accuracy sequence satisfies

$$\varepsilon_k > 0, \qquad \sum_{k=0}^{\infty} \varepsilon_k < \infty. \tag{24}$$

Such inexactness conditions are standard in KL-based analyses of nonconvex block coordinate methods. They can be enforced by running the inner BAPG routine to a prescribed tolerance $\varepsilon_k$ (e.g., by monitoring a stationarity residual), and choosing $\{\varepsilon_k\}$ to be summable. See, for instance, Attouch et al., 2013 for the abstract framework and Li et al., 2023 for Bregman-type transport updates in GW settings.

*Proof of Lemma 4.3.* Since each outer iterate $x^{(k)}$ is feasible, $\Phi(x^{(k)}) = \mathcal{L}(x^{(k)})$. Decompose the objective decrease across the three blocks:

$$\mathcal{L}(x^{(k)}) - \mathcal{L}(x^{(k+1)}) = (I) + (II) + (III), \tag{25}$$

where

$$(I) := \mathcal{L}(T^{(k)}, \alpha^{(k)}, \beta^{(k)}) - \mathcal{L}(T^{(k+1)}, \alpha^{(k)}, \beta^{(k)}),$$
$$(II) := \mathcal{L}(T^{(k+1)}, \alpha^{(k)}, \beta^{(k)}) - \mathcal{L}(T^{(k+1)}, \alpha^{(k+1)}, \beta^{(k)}),$$
$$(III) := \mathcal{L}(T^{(k+1)}, \alpha^{(k+1)}, \beta^{(k)}) - \mathcal{L}(T^{(k+1)}, \alpha^{(k+1)}, \beta^{(k+1)}).$$

For $(I)$, the inexact $T$-oracle gives:

$$(I) \geq c_T \left\| T^{(k+1)} - T^{(k)} \right\|_F^2 - e_k. \tag{26}$$

For $(II)$, the proximal subproblem (7) is $\delta$-strongly concave in $\alpha$ (equivalently, strongly convex after sign flip), hence the standard three-point inequality yields

$$(II) \geq \frac{\delta}{2} \left\| \alpha^{(k+1)} - \alpha^{(k)} \right\|_F^2. \tag{27}$$

Similarly,

$$(III) \geq \frac{\delta}{2} \left\| \beta^{(k+1)} - \beta^{(k)} \right\|_F^2. \tag{28}$$

Combining the three bounds and setting

$$\rho_1 := \min \left\{ c_T, \ \frac{\delta}{2} \right\}, \qquad E_k := e_k, \tag{29}$$

we obtain Lemma 4.3. □

*Proof of Lemma 4.4.* We bound each block of $\partial \Phi = \partial(\mathcal{L} + \delta_{\mathcal{D}})$.

**T-block.** By Assumption C.3,

$$\text{dist}\Big(0, \ \nabla_T \mathcal{L}(T^{(k+1)}, \alpha^{(k)}, \beta^{(k)}) + \mathcal{N}_{\Pi(\mu,\nu)}(T^{(k+1)})\Big) \tag{30}$$

$$\leq C_T \left\| T^{(k+1)} - T^{(k)} \right\|_F + \tilde{e}_k. \tag{31}$$

Using Lipschitz continuity of $\nabla_T \mathcal{L}$ on the compact set $\mathcal{D}$ (Assumption C.1),

$$\left\| \nabla_T \mathcal{L}(T^{(k+1)}, \alpha^{(k+1)}, \beta^{(k+1)}) - \nabla_T \mathcal{L}(T^{(k+1)}, \alpha^{(k)}, \beta^{(k)}) \right\| \tag{32}$$

$$\leq L_T \left\| (\alpha^{(k+1)}, \beta^{(k+1)}) - (\alpha^{(k)}, \beta^{(k)}) \right\|_F, \tag{33}$$

for some constant $L_T > 0$. Hence,

$$\text{dist}\Big(0, \ \partial_T \Phi(x^{(k+1)})\Big) \leq C_T \left\| T^{(k+1)} - T^{(k)} \right\|_F + \tag{34}$$

$$L_T \left\| (\alpha^{(k+1)}, \beta^{(k+1)}) - (\alpha^{(k)}, \beta^{(k)}) \right\|_F + \tilde{e}_k. \tag{35}$$

$\alpha$**-block.** The optimality condition of (7) implies

$$0 \in -G^\alpha + \delta \left( \alpha^{(k+1)} - \alpha^{(k)} \right) + \mathcal{N}_{\mathcal{A}(T^{(k+1)})}\left( \alpha^{(k+1)} \right), \tag{36}$$

where $\mathcal{N}_{\mathcal{A}(T)}$ is the normal cone. Since $G^\alpha$ equals $-\nabla_\alpha \mathcal{L}(T^{(k+1)}, \alpha^{(k+1)}, \beta^{(k)})$ by construction of the gain (it is the coefficient of $\alpha$ in $\mathcal{L}$), we obtain

$$\text{dist}\Big(0, \ \partial_\alpha \Phi(T^{(k+1)}, \alpha^{(k+1)}, \beta^{(k)})\Big) \leq \delta \left\| \alpha^{(k+1)} - \alpha^{(k)} \right\|_F. \tag{37}$$

A Lipschitz argument w.r.t. $\beta$ (compactness of $\mathcal{D}$) yields an additional term proportional to $\|\beta^{(k+1)} - \beta^{(k)}\|_F$, hence

$$\text{dist}\big(0, \partial_\alpha \Phi(x^{(k+1)})\big) \leq \delta \left\| \alpha^{(k+1)} - \alpha^{(k)} \right\|_F + \tag{38}$$

$$L_{\alpha\beta} \left\| \beta^{(k+1)} - \beta^{(k)} \right\|_F, \tag{39}$$

for some constant $L_{\alpha\beta} > 0$.

$\beta$**-block.** Similarly, from (8),

$$\text{dist}\big(0, \partial_\beta \Phi(x^{(k+1)})\big) \leq \delta \left\| \beta^{(k+1)} - \beta^{(k)} \right\|_F. \tag{40}$$

Combining (34), (38), and (40) and using $\|(a, b, c)\|_F \leq \|a\|_F + \|b\|_F + \|c\|_F$, we obtain (10) with a constant $\rho_2 > 0$ and $\varepsilon_k := \tilde{e}_k$, which is summable by Assumption C.3. □

## C.1. Experimental Details

### C.1.1. SYNTHETIC 2D GEOMETRIC MATCHING

In this section, we provide a comprehensive description of the synthetic data generation process and a detailed qualitative analysis of the matching results presented in the main text.

**Data Generation and Protocol.** To strictly evaluate robustness against non-rigid distortions and scale variations, we constructed a dataset based on various generic 2D manifolds. Specifically, we generated source point clouds $\mathcal{X} = \{x_i\}_{i=1}^n$ by sampling $n = 80$ points from distinct geometric primitives. The target point clouds $\mathcal{Y} = \{y_j\}_{j=1}^m$ were generated by applying a random rigid transformation combined with a scaling factor and local noise to the source points. The transformation model is defined as:

$$y_{\pi(i)} = sRx_i + t + \xi_i, \quad \forall i = 1, \ldots, n \tag{41}$$

where $R \in \mathbb{R}^{2 \times 2}$ is a random rotation matrix, $t \in \mathbb{R}^2$ is a random translation vector, and $s$ is a scaling factor sampled uniformly from $\mathcal{U}[0.9, 1.4]$. The term $\xi_i \sim \mathcal{N}(0, \sigma^2 I)$ introduces local Gaussian noise with $\sigma = 0.05$, which simulates non-rigid deformations and measurement errors, ensuring the task is not a simple rigid registration problem.

**Structured Outlier Simulation.** A critical challenge in our evaluation is the inclusion of *structured outliers*. We set the outlier ratio to approximately $20\%$ ($m \approx 1.2n$). Unlike random uniform noise, which is easily filtered by distance-based heuristics, structured outliers are arranged in coherent geometric patterns (e.g., clusters or linear features) positioned in proximity to the target inliers. These patterns are designed to maximize structural ambiguity, creating deceptive "local optima" for topology-based matchers. This setup rigorously tests the solver's ability to differentiate the true geometric signal from coherent background interference.

**Qualitative Analysis.** Figure 6 visualizes the matching results. The first column displays the input data configuration, highlighting the difficulty of determining the correct alignment due to the presence of outliers (red crosses). The subsequent columns show the learned coupling matrices $\Gamma$ by UGW, PGW, RGW, and our proposed Mask-Guided GW. Existing methods (UGW, PGW) struggle significantly, producing diffuse coupling matrices where probability mass is spread across off-diagonal elements. This indicates uncertainty and incorrect mappings caused by the interference of structured outliers. RGW improves sparsity but still retains noticeable noise. In contrast, **Mask-Guided GW** consistently recovers a sharp, clean diagonal structure across all scenarios. This demonstrates that our method effectively filters out structured outliers and accurately recovers the underlying geometric transformation.

### C.1.2. SUBGRAPH ALIGNMENT TASK

**Datasets** We evaluate the proposed method on four widely used graph benchmarks: **BZR** (Sutherland et al., 2003), **DHFR** (Sutherland et al., 2003), **Enzymes** (Borgwardt et al., 2005), and **Fingerprint** (Riesen & Bunke, 2008). The first three are bioinformatics datasets where graphs represent chemical compounds or protein tertiary structures; nodes denote atoms or secondary structure elements, and edges encode chemical bonds or spatial proximity. The **Fingerprint** dataset consists of graphs derived from biometric images, where nodes represent minutiae points. Across all datasets, node attributes play a crucial role in distinguishing functional or structural features.

**Experimental Settings**

**Subgraph Matching Task.** Unlike standard full-graph matching, we design a subgraph matching experiment to evaluate the robustness of GW methods against partial observability and size mismatch. For each graph $G$ in the dataset, we construct a *target graph* (subgraph) by randomly sampling $50\%$ of the nodes from $G$ (i.e., sampling ratio $\rho = 0.5$). The goal is to recover the correspondence between the nodes of the subgraph and their counterparts in the original graph.

**Attribute-based Geometry.** To challenge the solvers with feature-based alignment tasks, the distance matrices $D_X$ (for the original graph) and $D_Y$ (for the subgraph) are not derived from shortest-path variations on edges. Instead, they are constructed using the **node attributes**. Specifically, for any two nodes $i, j$, the distance $D_{ij}$ is computed as the Euclidean distance between their attribute vectors. This setup tests the methods' ability to align geometric structures defined by high-dimensional feature spaces rather than explicit connectivity.

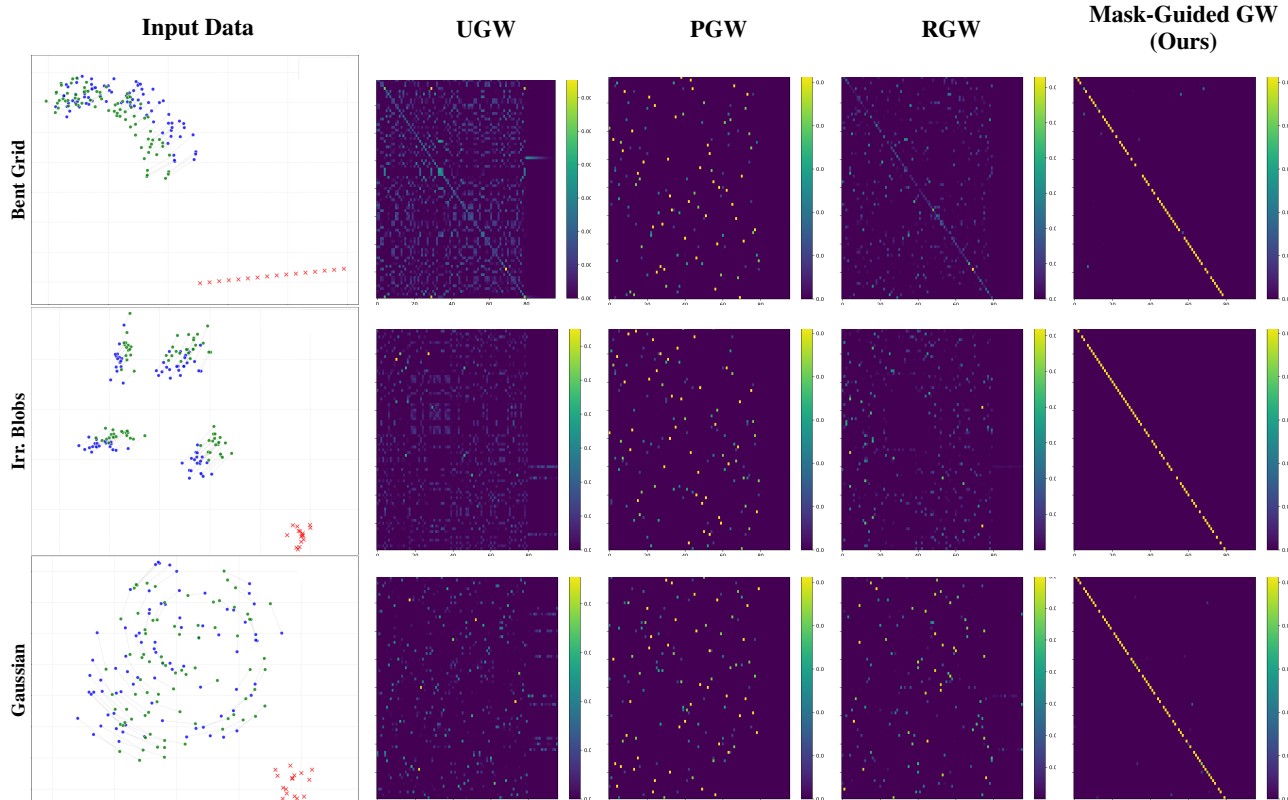

*Figure 6.* **Comparison of robustness across datasets.** The first column (**Input Data**, enlarged for visibility) visualizes the source (blue) and target (green/red) distributions. The subsequent columns display the coupling matrices from different methods. Our **Mask-Guided GW** (rightmost) consistently exhibits the cleanest diagonal structure.

## Implementation Details

**Initialization.** We utilize distance matrices, $D_X$ and $D_Y$, as the input distance matrices, with both source ($\mu$) and target ($\nu$) distributions set to uniform. For the baseline methods SpecGW, BPG, eBPG, BAPG, MD-SRGW, and RGW, we adhere to the configurations recommended in their respective original papers. Specifically, SpecGW employs the heat kernel $\exp(-L)$ derived from the normalized graph Laplacian $L$, while FW is implemented using the default settings in the PythonOT library. The transport plan is initialized uniformly as $\mathbf{1}_{n \times m}/nm$.

**Hyperparameter Tuning.** For methods requiring parameter selection, we perform grid searches to identify optimal configurations. For UGW, we tune the regularization parameter from $\{0.5, 0.2, \ldots, 0.001\}$ and the marginal penalty from $\{0.1, 0.01, 0.001\}$. For PGW, we report results based on the best retained mass ratio in the range $[0.1, 0.9]$. For RGW, we fix $\tau_1 = \tau_2 = 0.1$ and tune the marginal relaxation parameters from $\{0.05, \ldots, 0.5\}$ and step sizes from $\{0.01, \ldots, 1\}$. For our Mask-Guided GW, we fix the $\epsilon_1, \epsilon_2 = \epsilon$ where $\epsilon$ = sampling ratio.

**Evaluation Metrics:** We compute the standard matching accuracy ($Acc$), defined as the intersection over the size of the ground truth set ($|S_{gt} \cap S_{pred}|/|S_{gt}|$).

### C.2. Graph Edge Alignment under Structural Noise

In this section, we detail the experimental protocol designed to evaluate the robustness of Gromov-Wasserstein solvers against *structural noise* (also referred to as edge outliers or topological shortcuts). Unlike node-level outliers which can be handled by partial matching, structural noise corrupts the relational information itself, posing a fundamental challenge to quadratic assignment problems.

**Structural Noise Injection.** This experiment is built upon the Subgraph Alignment setting (Section C.1.2). Specifically, the source graph $\mathcal{G}_s$ is a subgraph sampled from the target graph $\mathcal{G}_t$ (which represents the complete graph), such that

*Table 2.* Comparison of matching accuracy (%) and runtime (s) on the **ENZYMES** dataset with varying subgraph ratios ($\rho$).

| Method | 0.1 | | 0.2 | | 0.3 | | 0.4 | | 0.5 | | 0.6 | |
|---|---|---|---|---|---|---|---|---|---|---|---|---|
| | Acc | Time | Acc | Time | Acc | Time | Acc | Time | Acc | Time | Acc | Time |
| FW | 17.77 | 0.39 | 10.82 | 1.21 | 7.72 | 1.42 | 7.94 | 3.05 | 7.18 | 0.48 | 8.13 | 3.10 |
| SpecGWL | 17.87 | 3.41 | 12.70 | 20.51 | 10.13 | 22.03 | 9.91 | 22.31 | 10.34 | 2.47 | 11.27 | 21.67 |
| eBPG | **22.07** | 3887.50 | 16.79 | 1.9e4 | 12.99 | 2.0e4 | 13.92 | 1.9e4 | 14.23 | 5938.37 | 15.85 | 2.0e4 |
| BPG | 9.22 | 10.54 | 9.08 | 59.19 | 10.49 | 65.71 | 12.99 | 74.19 | 14.26 | 17.25 | 16.86 | 85.93 |
| BAPG | 22.26 | 3.99 | 22.16 | 37.79 | 23.01 | 43.92 | 27.03 | 75.37 | 30.56 | 8.60 | 34.91 | 114.84 |
| MD-SRGW | 3.20 | 91.89 | 11.92 | 287.89 | 17.55 | 365.46 | 22.98 | 432.05 | 28.60 | 109.85 | 34.76 | 630.67 |
| UGW | 3.70 | 157.47 | 11.29 | 552.30 | 15.56 | 638.86 | 20.61 | 630.73 | 26.51 | 185.11 | 33.24 | 746.42 |
| PGW | 13.51 | 0.68 | 16.06 | 2.76 | 14.32 | 3.92 | 15.03 | 7.11 | 18.79 | 1.13 | 24.53 | 13.87 |
| RGW | 4.74 | 646.48 | 13.15 | 3214.74 | 21.72 | 3766.26 | 31.83 | 3888.03 | 41.62 | 1142.00 | 53.35 | 3560.18 |
| **Mask-Guided GW** | 17.91 | 1231.90 | **24.88** | 615.00 | **32.22** | 809.66 | **43.12** | 1111.02 | **52.28** | 77.08 | **57.49** | 1649.70 |

*Table 3.* Comparison of matching accuracy (%) and runtime (s) on the **Fingerprint** dataset with varying subgraph ratios ($\rho$). Mask-Guided GW demonstrates superior accuracy as the structural overlap increases.

| Method | $\rho = 0.2$ | | $\rho = 0.3$ | | $\rho = 0.4$ | | $\rho = 0.5$ | | $\rho = 0.6$ | |
|---|---|---|---|---|---|---|---|---|---|---|
| | Acc | Time | Acc | Time | Acc | Time | Acc | Time | Acc | Time |
| FW | 43.32 | 0.41 | 32.83 | 0.99 | 23.15 | 1.33 | 15.72 | 1.30 | 15.33 | 1.79 |
| SpecGWL | **47.48** | 1.95 | 36.57 | 4.35 | 31.71 | 6.17 | 19.39 | 5.86 | 17.25 | 7.20 |
| eBPG | 38.28 | 3119.24 | 38.30 | 11412.09 | 38.94 | 15466.42 | 37.46 | 14854.03 | 37.81 | 17625.34 |
| BPG | 22.59 | 10.53 | 29.70 | 28.42 | 29.32 | 44.84 | 31.17 | 46.90 | 34.62 | 47.88 |
| BAPG | 36.94 | 1.50 | **40.97** | 6.58 | 38.63 | 9.57 | 40.16 | 8.81 | 42.93 | 11.53 |
| MD-SRGW | 20.06 | 91.97 | 25.62 | 208.06 | 24.26 | 267.19 | 24.27 | 240.53 | 26.41 | 253.17 |
| UGW | 12.15 | 287.61 | 32.00 | 1617.33 | 39.97 | 2714.99 | 43.29 | 1904.25 | 43.61 | 1368.62 |
| PGW | 28.47 | 0.94 | 40.91 | 2.19 | 35.69 | 2.90 | 35.68 | 2.30 | 33.46 | 2.65 |
| RGW | 16.59 | 568.79 | 26.59 | 1899.41 | 25.37 | 2474.15 | 25.85 | 1704.16 | 26.31 | 1723.16 |
| **Mask-Guided GW** | 26.75 | 127.80 | 38.76 | 76.63 | **46.29** | 109.10 | **47.51** | 233.78 | **51.74** | 479.55 |

$\mathcal{V}_s \subset \mathcal{V}_t$. We then inject noise into the target graph $\mathcal{G}_t$ to simulate geometric defects in the background structure. We rely on *edge weight perturbation* to generate this noise. Let $\eta \in [0, 1]$ denote the noise ratio. We randomly select a subset of edges $\mathcal{E}_{noise} \subset \mathcal{E}_t$ in the target graph such that $|\mathcal{E}_{noise}| = \lfloor \eta \cdot |\mathcal{E}_t| \rfloor$. For these edges, we corrupt their weights (distances) with random noise. This setup challenges the solver to align a partial structure (source) to a noisy complete structure (target), testing robustness against both occlusion and geometric distortion.

**Evaluation Metric: Edge Accuracy.** Standard node matching accuracy is insufficient for assessing topological preservation, as it does not explicitly penalize the disruption of graph structures. To rigorously evaluate the recovery of the geometric skeleton, we employ the Edge Matching Accuracy (EMA). This metric quantifies the proportion of source edges that are correctly mapped to valid target edges based on the transport plan. Given the computed transport plan $P \in \mathbb{R}^{n \times m}$ and the ground-truth node correspondence $\pi^*$, we define the matching score for a pair of edges $(i, j) \in \mathcal{E}_s$ and $(k, l) \in \mathcal{E}_t$ as the joint probability mass transported between their endpoints. The predicted target edge for a source edge $(i, j)$ is obtained by maximizing this score:

$$\hat{e}_t(i, j) = \arg \max_{(k,l) \in \mathcal{E}_t} (P_{ik} P_{jl} + P_{il} P_{jk}), \tag{42}$$

where the term $P_{ik} P_{jl}$ represents the probability of mapping node $i$ to $k$ and node $j$ to $l$ (accounting for undirected edges via symmetry). The EMA is then calculated as the percentage of source edges whose predicted counterparts align with the ground truth:

$$\text{EMA} = \frac{1}{|\mathcal{E}_s|} \sum_{(i,j) \in \mathcal{E}_s} \mathbb{I} \left[ \hat{e}_t(i, j) = (\pi^*(i), \pi^*(j)) \right], \tag{43}$$

where $\mathbb{I}[\cdot]$ denotes the indicator function. A higher EMA indicates that the method successfully filters structural noise and preserves the underlying connectivity of the graph.

*Table 4.* Comparison of matching accuracy (%) and runtime (s) on the **DHFR** dataset. This dataset proves more challenging at low ratios, but Mask-Guided GW achieves significant gains as $\rho$ increases.

| Method | $\rho = 0.2$ | | $\rho = 0.3$ | | $\rho = 0.4$ | | $\rho = 0.5$ | | $\rho = 0.6$ | | $\rho = 0.7$ | |
|---|---|---|---|---|---|---|---|---|---|---|---|---|
| | Acc | Time | Acc | Time | Acc | Time | Acc | Time | Acc | Time | Acc | Time |
| FW | 10.60 | 1.12 | 8.90 | 1.58 | 8.40 | 1.84 | 10.17 | 1.34 | 10.59 | 2.65 | 11.70 | 3.16 |
| SpecGWL | 9.38 | 9.24 | 8.16 | 9.50 | 6.93 | 10.04 | 8.50 | 7.19 | 11.19 | 11.68 | 11.84 | 11.94 |
| eBPG | **12.66** | 1.6e4 | **8.96** | 1.7e4 | 7.74 | 1.5e4 | 8.28 | 1.3e4 | 11.42 | 1.5e4 | 10.65 | 1.4e4 |
| BPG | 7.73 | 32.79 | 6.62 | 36.27 | 7.07 | 41.59 | 10.17 | 38.52 | 18.65 | 59.10 | 25.43 | 59.65 |
| BAPG | 4.20 | 22.07 | 5.21 | 24.77 | 7.68 | 26.09 | 12.13 | 23.96 | 25.72 | 32.90 | 32.54 | 35.01 |
| MD-SRGW | 3.63 | 270.07 | 4.76 | 261.01 | 5.39 | 282.16 | 12.03 | 261.42 | 22.39 | 353.39 | 28.54 | 396.16 |
| UGW | 5.82 | 424.34 | 5.98 | 412.68 | 8.83 | 445.02 | 14.78 | 387.92 | 40.79 | 506.39 | 51.54 | 485.81 |
| PGW | 5.02 | 3.57 | 4.26 | 4.75 | 5.38 | 5.61 | 11.15 | 4.67 | 23.87 | 9.52 | 30.09 | 9.71 |
| RGW | 4.54 | 2693.03 | 4.34 | 3648.09 | 9.29 | 4675.10 | 23.01 | 4945.09 | 48.20 | 5942.77 | 57.36 | 6328.60 |
| **Mask-Guided GW** | 4.01 | 85.11 | 6.59 | 55.17 | **17.30** | 102.48 | **34.08** | 265.06 | **59.61** | 366.27 | **57.52** | 321.29 |

**Hypothesis.** We hypothesize that standard GW will be sensitive to the global distortion of pairwise distances caused by noise. Partial GW (PGW) may struggle as it cannot selectively filter edge costs without dropping nodes. Our method is designed to identify these distorted interactions via the auxiliary variables $\alpha, \beta$ and "mute" the high-error edge costs, recovering the true node correspondence.

*Table 5.* **Performance on Structural Noise Task ($\eta = 0.1$).** We report the Node Accuracy ($N_{acc}$, %), Soft Edge Accuracy ($E_{acc}$, %), and Runtime ($T$, seconds) across four datasets. Our Mask-Guided GW consistently achieves the best topological recovery ($E_{acc}$) and efficiency ($T$) among robust solvers. Best results in **bold**.

| Method | BZR | | | Enzymes | | | DHFR | | | Fingerprint | | |
|---|---|---|---|---|---|---|---|---|---|---|---|---|
| | $N_{acc}$ | $E_{acc}$ | $T$ | $N_{acc}$ | $E_{acc}$ | $T$ | $N_{acc}$ | $E_{acc}$ | $T$ | $N_{acc}$ | $E_{acc}$ | $T$ |
| *Balanced / Standard Methods* | | | | | | | | | | | | |
| FW | 13.6 | 2.2 | 0.7 | 7.6 | 1.6 | 1.0 | 9.8 | 1.4 | 1.4 | 15.7 | 3.7 | 1.5 |
| SpecGWL | 11.4 | 1.4 | 2.9 | 9.7 | 2.0 | 5.5 | 8.3 | 1.0 | 8.2 | 19.4 | 14.5 | 6.7 |
| eBPG | 13.5 | 2.1 | 5763 | 13.8 | 3.1 | 10384 | 8.1 | 1.4 | 11969 | 37.3 | 31.3 | 16701 |
| BPG | 17.9 | 3.8 | 17.9 | 13.1 | 2.7 | 31.9 | 9.9 | 2.1 | 36.6 | 31.0 | 20.8 | 52.3 |
| BAPG | 18.8 | 4.8 | 8.9 | 28.8 | 11.0 | 16.8 | 12.1 | 3.0 | 21.3 | 40.1 | 37.3 | 9.7 |
| *Unbalanced / Robust Methods* | | | | | | | | | | | | |
| MD-SRGW | 23.6 | 14.3 | 99 | 27.9 | 15.1 | 195 | 11.3 | 6.5 | 231 | 24.1 | 14.5 | 262 |
| UGW | 30.1 | 17.1 | 205 | 25.7 | 11.0 | 340 | 14.8 | 6.3 | 407 | 43.3 | 47.3 | 2078 |
| PGW | 14.5 | 5.6 | 1.6 | 18.6 | 7.2 | 4.5 | 10.7 | 4.7 | 4.5 | 35.8 | 29.6 | 2.6 |
| RGW | 43.3 | 36.9 | 1568 | 38.3 | 26.2 | 2609 | 21.6 | 18.7 | 5249 | 25.5 | 16.6 | 1981 |
| **Mask-Guided GW (Ours)** | **49.6** | **43.2** | 130 | **44.6** | **30.1** | 370 | **35.0** | **27.0** | 448 | **50.9** | **60.8** | 392 |

### C.2.1. PERFORMANCE ON SUBGRAPH ALIGNMENT

We report the quantitative matching results on the Enzymes (Table 2), Fingerprint (Table 3), and DHFR (Table 4) datasets. The analysis focuses on matching accuracy and computational efficiency under varying degrees of partial observability (sampling ratio $\rho$).

**Accuracy under Partial Observability.** Across all three datasets, **Mask-Guided GW** consistently outperforms baseline methods, particularly as the structural overlap increases ($\rho \geq 0.4$).

- **Enzymes (Table 2):** Our method achieves a dominant performance. At $\rho = 0.5$, we surpass the second-best method (RGW) by a significant margin (52.28% vs. 41.62%). This indicates that for bio-molecular graphs with complex attribute spaces, our method better preserves the global topology.

- **Fingerprint (Table 3):** This dataset represents geometric graphs where node positions are critical. While spectral methods (SpecGWL) show strength at very low overlaps ($\rho = 0.2$), they degrade rapidly as $\rho$ increases. Our method

demonstrates robust scaling behavior, achieving the highest accuracy of $51.74\%$ at $\rho = 0.6$, significantly outperforming UGW ($43.61\%$) and PGW ($33.46\%$).

- **DHFR (Table 4):** On this challenging dataset, where most methods struggle to exceed $20\%$ accuracy at low ratios, our method shows superior adaptability. At $\rho = 0.6$, Mask-Guided GW achieves $59.61\%$, effectively solving the alignment problem where balanced methods (FW, SpecGWL) fail completely ($< 12\%$).

**Computational Efficiency.** A critical advantage of Mask-Guided GW is its balance between accuracy and runtime. Robust solvers like eBPG and RGW rely on computationally expensive routines. For instance, on the Enzymes dataset ($\rho = 0.6$), RGW requires over 3,500 seconds, and eBPG exceeds 20,000 seconds. In stark contrast, Mask-Guided GW converges in approximately 1,600 seconds while delivering higher accuracy. On the Fingerprint dataset ($\rho = 0.5$), our method is $7\times$ faster than RGW ($233s$ vs. $1704s$). This efficiency gain is attributed to the mask-guided scheme, which accelerates convergence by pruning the search space of the transport plan, avoiding the need for exhaustive relaxation used in RGW.

### C.2.2. TOPOLOGICAL RECOVERY UNDER STRUCTURAL NOISE

Table 5 presents the evaluation of edge alignment under structural noise ($\eta = 0.1$). This experiment explicitly tests whether the solvers can distinguish true topological connections from corrupted edges.

**Superior Edge Recovery.** The Edge Accuracy ($E_{acc}$) serves as a direct proxy for topological fidelity. Standard methods (FW, SpecGWL) and even some robust methods (PGW) collapse under structural noise, yielding $E_{acc} < 10\%$ on most datasets. This confirms our hypothesis that global cost functions are easily distorted by edge perturbations. **Mask-Guided GW** achieves the highest $E_{acc}$ across all benchmarks. On the **BZR** dataset, we achieve $43.2\%$ edge accuracy compared to $36.9\%$ for RGW. Notably, on the **Fingerprint** dataset, our method reaches $60.8\%$ edge accuracy, surpassing UGW ($47.3\%$) by a large margin. The gap between Node Accuracy ($N_{acc}$) and Edge Accuracy ($E_{acc}$) is minimal for our method compared to others, suggesting that Mask-Guided GW does not merely match nodes based on local descriptors but successfully aligns the underlying graph skeleton, effectively "muting" the noisy edges during the optimization process.

### C.3. Additional Benchmark Results

**General graph structures.** To evaluate whether the observed gains are specific to attribute-based Euclidean distance matrices, we also run subgraph alignment on ENZYMES using unweighted shortest-path matrices. The protocol follows the main subgraph-matching setting, including the same graph-pair construction, hyperparameter selection, and matching-accuracy metric; no additional structural noise is injected. Table 6 shows that Mask-Guided GW remains the strongest method, improving the mean accuracy from $0.4554$ for the best baseline to $0.5658$.

*Table 6.* ENZYMES subgraph benchmark using shortest-path matrices.

| Method | Mean Acc | Std | RowDev Mean | RowDev Std | ColDev Mean | ColDev Std |
|---|---|---|---|---|---|---|
| Mask-Guided GW | **0.5658** | 0.2403 | 2.1943e-01 | 1.0360e-01 | 1.0423e-03 | 4.5404e-03 |
| FW | 0.1993 | 0.0979 | 1.9275e-08 | 8.9565e-09 | 1.6674e-08 | 2.4737e-08 |
| BAPGcpu | 0.0767 | 0.0410 | 2.3957e-01 | 2.2079e-02 | 7.0221e-08 | 1.5164e-08 |
| BPG | 0.2508 | 0.1576 | 1.0825e-16 | 3.5494e-17 | 9.5263e-06 | 2.6769e-06 |
| MD-SRGW | 0.2775 | 0.2065 | 4.8787e-08 | 1.1332e-08 | 4.0091e-01 | 2.2625e-01 |
| RGW | 0.4554 | 0.2617 | 2.4501e-01 | 4.5295e-02 | 4.5825e-01 | 1.0917e-01 |
| PGW | 0.3003 | 0.2552 | 1.0000e-01 | 8.4547e-08 | 1.0000e-01 | 6.4794e-08 |
| RGDL_md | 0.0555 | 0.0463 | 1.4849e-16 | 3.4075e-17 | 2.4587e-08 | 1.7519e-08 |

**Pairwise-perturbation baseline and marginal deviations.** Table 7 reports the ENZYMES subgraph benchmark with the same protocol used in the main text. The distance matrices are constructed from node-attribute Euclidean distances, and the matching direction is subgraph-to-full graph: the sampled subgraph is used as the source graph and the complete graph as the target graph. The comparison also reports marginal deviations. The dual-mask mechanism improves matching accuracy

while its feasibility profile remains comparable to the underlying BAPG-style transport update; in particular, it does not introduce the large column deviation observed in unbalanced alternatives.

*Table 7.* ENZYMES subgraph benchmark with marginal deviation statistics.

| Method | Mean Acc | Std | RowDev Mean | RowDev Std | ColDev Mean | ColDev Std |
|---|---|---|---|---|---|---|
| Mask-Guided GW | **0.5238** | 0.3418 | 3.8916e-01 | 1.4580e-01 | 1.0791e-03 | 1.6493e-02 |
| FW | 0.0717 | 0.0943 | 8.1923e-09 | 1.4268e-08 | 9.5360e-09 | 2.5343e-08 |
| BAPGcpu | 0.2116 | 0.1669 | 1.9584e-01 | 9.5508e-02 | 5.3105e-08 | 1.2102e-08 |
| BPG | 0.1449 | 0.1359 | 1.1268e-16 | 3.9628e-17 | 7.0680e-06 | 3.1675e-06 |
| eBPG | 0.1432 | 0.1258 | 6.8877e-08 | 2.1119e-08 | 1.3552e-07 | 2.0093e-06 |
| SpecGWL | 0.1027 | 0.1241 | 5.6577e-09 | 9.0274e-09 | 9.4047e-09 | 2.1487e-08 |
| MD-SRGW | 0.2854 | 0.2750 | 4.6215e-08 | 1.0784e-08 | 6.3893e-01 | 2.1646e-01 |
| RGW | 0.4157 | 0.3722 | 2.3932e-01 | 4.1406e-02 | 6.0549e-01 | 1.2466e-01 |
| PGW | 0.1893 | 0.2165 | 1.0000e-01 | 7.2199e-08 | 1.0000e-01 | 6.4428e-08 |
| RGDL_md | 0.1092 | 0.0859 | 1.4407e-16 | 4.8137e-17 | 9.7215e-09 | 1.7818e-08 |

**Shape matching and graph clustering.** We further evaluate ModelNet10 subgraph matching by randomly selecting 100 shapes, sampling 50 points from each shape, and using a subgraph ratio of $\rho = 0.5$. Table 8 reports Top-1 and Top-5 accuracy, where Mask-Guided GW achieves the best result. Table 9 reports average runtime over different graph sizes on ModelNet, where the graph sizes 50, 100, 200, and 500 denote the number of nodes. The runtime experiment uses the same random-shape sampling protocol, keeps hyperparameters fixed across graph sizes, and follows the same hardware setting as the main experiments. The runtime trend is consistent with the per-iteration complexity: the dominant GW-type contraction has the same leading order as BAPG, while mask updates add sorting overhead. We also evaluate ENZYMES graph clustering by using each method as a GW-based graph-distance calculator in a K-means evaluation; Table 10 shows that the learned structural distance is useful beyond node-level alignment.

*Table 8.* ModelNet10 subgraph matching.

| Method | Top-1 Acc | Top-5 Acc |
|---|---|---|
| UGW | 0.5104 | 0.8802 |
| PGW | 0.4844 | 0.6979 |
| RGW | 0.5365 | 0.7812 |
| MD-SRGW | 0.3021 | 0.6510 |
| Mask-Guided GW | **0.6979** | **0.9323** |

*Table 9.* Average runtime (seconds) on ModelNet across graph sizes.

| Method | 50 | 100 | 200 | 500 |
|---|---|---|---|---|
| Mask-Guided GW | **0.39** | **0.53** | **0.78** | **0.87** |
| RGW | 4.66 | 18.51 | 23.38 | 64.42 |
| UGW | 0.58 | 0.84 | 1.07 | 1.25 |

*Table 10.* Graph clustering accuracy on ENZYMES using K-means with GW-based graph distances.

| Method | Accuracy |
|---|---|
| Mask-Guided GW | **0.7917** |
| UGW | 0.7396 |
| PGW | 0.4792 |
| RGW | 0.2188 |

## C.4. Sensitivity and Feasibility Analysis

**Budget selection.** The budget $\epsilon$ controls the total amount of structural interaction mass that can be filtered. In subgraph matching, we tie $\epsilon$ to the known sampling ratio, yielding a simple default tied to the expected retained structure. When the corruption level is unavailable, a lightweight sweep around an anticipated retained-structure ratio can be used; alternatively, one may select $\epsilon$ using an unsupervised proxy such as reconstruction consistency under the learned coupling. The extreme cases are also informative: when $\epsilon = 0$, the masks are forced to vanish and the objective reduces to standard GW; when $\epsilon$ is too large, the optimizer can mask too many interactions, leading to a near-zero cost that no longer carries reliable alignment information.

**Penalty sensitivity.** Table 11 studies the sensitivity of Mask-Guided GW to the BAPG penalty parameter $\lambda$ on ENZYMES. The method remains competitive over a moderate range and peaks at $\lambda = 0.01$. For comparison, Table 12 reports the sensitivity of RGW to its key parameter $\eta$ under the same benchmark family.

*Table 11.* Sensitivity of Mask-Guided GW to $\lambda$ on ENZYMES.

| $\lambda$ | Mean Acc | Std |
|---|---|---|
| 0.005 | 0.5347 | 0.3130 |
| 0.01 | **0.5657** | 0.3232 |
| 0.05 | 0.5233 | 0.3035 |
| 0.1 | 0.4376 | 0.3229 |
| 0.5 | 0.2957 | 0.2453 |

*Table 12.* Sensitivity of RGW to $\eta$ on ENZYMES.

| $\eta$ | Mean Acc | Std |
|---|---|---|
| 0.005 | **0.4113** | 0.3704 |
| 0.01 | 0.3891 | 0.3521 |
| 0.05 | 0.3975 | 0.3640 |
| 0.1 | 0.3544 | 0.3323 |
| 0.5 | 0.2764 | 0.2966 |

**Feasibility profile.** The marginal statistics in Tables 6 and 7 show that the dual-mask mechanism does not by itself relax the transport marginal constraints. The observed row deviations are tied to the BAPG-style transport update and the unequal-size subgraph setting, while the column deviation of Mask-Guided GW remains small compared with unbalanced baselines such as RGW and MD-SRGW.

## C.5. Further Theoretical Properties

**Exact greedy mask updates.** The convergence theorem in Section 4 is stated for proximalized mask updates because the proximal term gives a direct sufficient-decrease inequality on the moving capped-simplex feasible sets. The exact greedy solver used in practice instead solves the continuous-knapsack mask subproblems exactly. We now give a local version showing that the same two KL ingredients, sufficient decrease and relative error, still hold for the exact greedy update once the moving feasible sets and the greedy active sets have stabilized.

Let $a = \mathrm{vec}(\alpha)$, $b = \mathrm{vec}(\beta)$, and $t = \mathrm{vec}(T)$. Define the capped-simplex feasible sets

$$\mathcal{A}(T) := \{a \in \mathbb{R}^{nm} : 0 \leq a \leq \mathrm{vec}(T), \ \mathbf{1}^\top a \leq \epsilon_1\},$$
$$\mathcal{B}(T) := \{b \in \mathbb{R}^{nm} : 0 \leq b \leq \mathrm{vec}(T), \ \mathbf{1}^\top b \leq \epsilon_2\}. \tag{44}$$

At iteration $k$, the exact greedy updates are equivalently written as

$$a^{(k+1)} \in \arg \max_{a \in \mathcal{A}(T^{(k+1)})} \langle g_\alpha^{(k)}, a \rangle,$$
$$b^{(k+1)} \in \arg \max_{b \in \mathcal{B}(T^{(k+1)})} \langle g_\beta^{(k)}, b \rangle, \tag{45}$$

where $g_\alpha^{(k)} = \mathrm{vec}(G^\alpha)$ and $g_\beta^{(k)} = \mathrm{vec}(G^\beta)$ are the gain vectors used by Algorithm 2.

**Assumption C.4** (Local stability and strict greedy gap). There exists an index $K$ such that for all $k \geq K$ the following hold. First, the previous masks remain feasible for the new capped-simplex sets:

$$a^{(k)} \in \mathcal{A}(T^{(k+1)}), \qquad b^{(k)} \in \mathcal{B}(T^{(k+1)}). \tag{46}$$

Second, each greedy knapsack solution has a clearly separated active-set structure. For the $\alpha$-update, there are disjoint sets $S_\alpha^{(k)}, P_\alpha^{(k)}, Z_\alpha^{(k)}$ (saturated, pivot, and zero coordinates) with $|P_\alpha^{(k)}| \leq 1$, a threshold $\tau_\alpha^{(k)} \geq \underline{\tau}_\alpha > 0$, and a gap $\gamma_\alpha > 0$ such that, with $t^{(k+1)} = \mathrm{vec}(T^{(k+1)})$,

$$a_i^{(k+1)} = t_i^{(k+1)} \ (i \in S_\alpha^{(k)}), \qquad a_j^{(k+1)} = 0 \ (j \in Z_\alpha^{(k)}), \tag{47}$$

the budget is active, $\mathbf{1}^\top a^{(k+1)} = \epsilon_1$, and

$$
\begin{aligned}
g_{\alpha,i}^{(k)} &\geq \tau_\alpha^{(k)} + \gamma_\alpha & (i \in S_\alpha^{(k)}), \\
g_{\alpha,j}^{(k)} &\leq \tau_\alpha^{(k)} - \gamma_\alpha & (j \in Z_\alpha^{(k)}), \\
g_{\alpha,p}^{(k)} &= \tau_\alpha^{(k)} & (P_\alpha^{(k)} = \{p\}).
\end{aligned} \tag{48}
$$

The $\beta$-update satisfies the same conditions with constants $\underline{\tau}_\beta > 0$ and $\gamma_\beta > 0$.

Assumption C.4 is a mild local condition rather than a consequence of the proximal analysis above. The feasibility part only requires that, once $T^{(k)}$ has settled into a regime of small changes, the previous mask is still a valid point of the next capped-simplex set. The strict-gap part is the standard non-degeneracy condition for greedy linear programs; it holds for almost every gain configuration, since exact ties between the threshold and a gain value form a measure-zero set under any continuous perturbation. In practice, deterministic tie-breaking picks one of the equivalent greedy optima without changing the greedy objective value, and a light projection onto the current capped-simplex set can be used as a feasibility safeguard if needed.

**Proposition C.5** (Sufficient decrease for exact greedy masks). *Under Assumptions C.1–C.3 and Assumption C.4, there exist constants $\kappa_g > 0$ and $K$ such that for all $k \geq K$,*

$$\Phi(x^{(k)}) - \Phi(x^{(k+1)}) \geq \kappa_g \|x^{(k+1)} - x^{(k)}\|_F^2 - \varepsilon_k. \tag{49}$$

*Proof.* As in Lemma 4.3, decompose the objective decrease into

$$\mathcal{L}(x^{(k)}) - \mathcal{L}(x^{(k+1)}) = (I) + (II) + (III), \tag{50}$$

where $(I)$ is the $T$-block decrease, $(II)$ is the $\alpha$-block decrease, and $(III)$ is the $\beta$-block decrease. Assumption C.3 gives

$$(I) \geq c_T \|T^{(k+1)} - T^{(k)}\|_F^2 - \varepsilon_k. \tag{51}$$

It remains to replace the proximal strong-concavity argument for $(II)$ and $(III)$ with an error bound for the exact greedy linear program.

We do the $\alpha$ case; the $\beta$ case is the same. Let

$$t := \text{vec}(T^{(k+1)}), \quad a^* := a^{(k+1)}, \quad a := a^{(k)}, \quad g := g_\alpha^{(k)}. \tag{52}$$

Since $\mathcal{L}$ is affine in $\alpha$ when $T$ and $\beta$ are fixed,

$$(II) = \langle g, a^* - a \rangle. \tag{53}$$

By Assumption C.4, $a \in \mathcal{A}(T^{(k+1)})$, so $a$ is a valid reference point for the current greedy subproblem. Let

$$A_S := \sum_{i \in S_\alpha^{(k)}} (t_i - a_i), \qquad A_Z := \sum_{j \in Z_\alpha^{(k)}} a_j, \qquad s := \epsilon_1 - \mathbf{1}^\top a \geq 0. \tag{54}$$

Because $\mathbf{1}^\top a^* = \epsilon_1$,

$$\langle g, a^* - a \rangle = \langle g - \tau_\alpha^{(k)} \mathbf{1}, a^* - a \rangle + \tau_\alpha^{(k)} s. \tag{55}$$

For $i \in S_\alpha^{(k)}$, we have $a_i^* = t_i$ and $g_i - \tau_\alpha^{(k)} \geq \gamma_\alpha$, so the contribution is at least $\gamma_\alpha(t_i - a_i)$. For $j \in Z_\alpha^{(k)}$, we have $a_j^* = 0$ and $\tau_\alpha^{(k)} - g_j \geq \gamma_\alpha$, so the contribution is at least $\gamma_\alpha a_j$. The pivot coordinate, if any, has $g_p = \tau_\alpha^{(k)}$, and so contributes zero after subtracting the threshold. Hence

$$\langle g, a^* - a \rangle \geq \gamma_\alpha(A_S + A_Z) + \underline{\tau}_\alpha s. \tag{56}$$

We next relate this quantity to the step size. If there is no pivot coordinate, then $\|a^* - a\|_1 = A_S + A_Z$. If $P_\alpha^{(k)} = \{p\}$, then

$$a_p^* - a_p = (\epsilon_1 - \mathbf{1}^\top a) - \sum_{i \in S_\alpha^{(k)}} (t_i - a_i) + \sum_{j \in Z_\alpha^{(k)}} a_j, \tag{57}$$

and therefore $|a_p^* - a_p| \leq s + A_S + A_Z$. In both cases,

$$\|a^* - a\|_1 \leq 2(A_S + A_Z) + s. \tag{58}$$

Combining this bound with (56) gives

$$\langle g, a^* - a \rangle \geq c_\alpha \|a^* - a\|_1, \qquad c_\alpha := \min\{\gamma_\alpha/2, \underline{\tau}_\alpha\}. \tag{59}$$

Since $0 \leq a_i, a_i^* \leq t_i \leq 1$, we have $\|a^* - a\|_\infty \leq 1$ and therefore $\|a^* - a\|_F^2 \leq \|a^* - a\|_\infty \cdot \|a^* - a\|_1 \leq \|a^* - a\|_1$, which gives

$$(II) \geq c_\alpha \|\alpha^{(k+1)} - \alpha^{(k)}\|_F^2. \tag{60}$$

The same argument gives

$$(III) \geq c_\beta \|\beta^{(k+1)} - \beta^{(k)}\|_F^2 \tag{61}$$

for some $c_\beta > 0$. Taking $\kappa_g = \min\{c_T, c_\alpha, c_\beta\}$ proves the claim. $\qquad \square$

**Proposition C.6** (Relative error for exact greedy masks). *Under the same conditions as Proposition C.5, there exist constants $\kappa_g' > 0$ and $K$ such that for all $k \geq K$,*

$$\text{dist}\big(0, \partial\Phi(x^{(k+1)})\big) \leq \kappa_g' \|x^{(k+1)} - x^{(k)}\|_F + \varepsilon_k. \tag{62}$$

*Proof.* The $T$-block is the same as in Lemma 4.4: the inexact $T$-oracle and Lipschitz continuity of $\nabla_T \mathcal{L}$ on the compact feasible set give

$$\text{dist}\big(0, \partial_T \Phi(x^{(k+1)})\big) \leq C_T \|T^{(k+1)} - T^{(k)}\|_F + L_T \|(\alpha^{(k+1)}, \beta^{(k+1)}) - (\alpha^{(k)}, \beta^{(k)})\|_F + \varepsilon_k. \tag{63}$$

For the $\alpha$-block, exact optimality of the greedy continuous-knapsack solution gives the first-order condition

$$0 \in -g_\alpha^{(k)} + \mathcal{N}_{\mathcal{A}(T^{(k+1)})}\left(a^{(k+1)}\right). \tag{64}$$

Recall that the gain vector satisfies $g_\alpha^{(k)} = -\mathrm{vec}(\nabla_\alpha \mathcal{L}(T^{(k+1)}, \alpha^{(k+1)}, \beta^{(k)}))$, and that the capped-simplex set $\mathcal{A}(T^{(k+1)})$ depends only on $T^{(k+1)}$ and not on $\beta$. So the $\alpha$-block subgradient residual is zero at the intermediate point $(T^{(k+1)}, \alpha^{(k+1)}, \beta^{(k)})$. Lipschitz continuity of $\nabla_\alpha \mathcal{L}$ in $\beta$ then gives

$$\mathrm{dist}\left(0, \partial_\alpha \Phi(x^{(k+1)})\right) \leq L_{\alpha\beta} \|\beta^{(k+1)} - \beta^{(k)}\|_F. \tag{65}$$

For the $\beta$-block, $\beta^{(k+1)}$ is computed last in the BCD sweep, so the same exact-optimality argument at $(T^{(k+1)}, \alpha^{(k+1)}, \beta^{(k+1)})$ gives

$$\mathrm{dist}\left(0, \partial_\beta \Phi(x^{(k+1)})\right) = 0. \tag{66}$$

Combining the three block bounds, as in the proof of Lemma 4.4, gives the relative-error inequality. $\square$

Propositions C.5 and C.6 show that, after a finite warm-up phase, exact greedy mask updates satisfy the same two KL ingredients used in the proof of Theorem 4.1. Dropping a finite prefix does not affect finite-length convergence or the limit point, so the KL convergence argument carries over locally to the exact greedy implementation under Assumption C.4. The extra assumption is needed because the feasible mask sets move with $T$; without the feasibility-preserving condition, the previous mask may stop being a valid reference point after the $T$-step.

**Objective-level perturbation bound.** Let $A_{ij,kl} := L(C_{ij}, D_{kl})$ be the clean interaction tensor and let $\widetilde{A} = A + \Delta$ be a perturbed version with entrywise change $\Delta_{ij,kl}$. Define

$$\Phi_A(T, \alpha, \beta) := \sum_{i,j,k,l} A_{ij,kl}(T_{ik} - \alpha_{ik})(T_{jl} - \beta_{jl}). \tag{67}$$

**Proposition C.7** (Mask-attenuated perturbation sensitivity)**.** *For any* $(T, \alpha, \beta) \in \mathcal{D}$,

$$\left|\Phi_{\widetilde{A}}(T, \alpha, \beta) - \Phi_A(T, \alpha, \beta)\right| \leq \|\Delta\|_\infty (1 - \|\alpha\|_1)(1 - \|\beta\|_1). \tag{68}$$

*Proof.* By bilinearity of $\Phi_\bullet(T, \alpha, \beta)$ in the interaction tensor,

$$\Phi_{\widetilde{A}}(T, \alpha, \beta) - \Phi_A(T, \alpha, \beta) = \sum_{i,j,k,l} \Delta_{ij,kl}(T_{ik} - \alpha_{ik})(T_{jl} - \beta_{jl}). \tag{69}$$

Feasibility $(T, \alpha, \beta) \in \mathcal{D}$ forces $0 \leq \alpha \leq T$ and $0 \leq \beta \leq T$ entrywise, so $T_{ik} - \alpha_{ik} \geq 0$ and $T_{jl} - \beta_{jl} \geq 0$ for all indices. Using the triangle inequality and the entrywise bound $|\Delta_{ij,kl}| \leq \|\Delta\|_\infty$,

$$\begin{aligned}
\left|\Phi_{\widetilde{A}} - \Phi_A\right| &\leq \|\Delta\|_\infty \sum_{i,j,k,l} (T_{ik} - \alpha_{ik})(T_{jl} - \beta_{jl}) \\
&= \|\Delta\|_\infty \left(\sum_{i,k}(T_{ik} - \alpha_{ik})\right)\left(\sum_{j,l}(T_{jl} - \beta_{jl})\right),
\end{aligned} \tag{70}$$

where the second equality splits the rank-one product over the independent index pairs $(i, k)$ and $(j, l)$. Since $T \in \Pi(\mu, \nu)$ has total mass $\sum_{i,k} T_{ik} = \langle \mathbf{1}, \mu \rangle = 1$, and $\alpha \geq 0$ gives $\|\alpha\|_1 = \sum_{i,k} \alpha_{ik}$,

$$\sum_{i,k}(T_{ik} - \alpha_{ik}) = 1 - \|\alpha\|_1, \qquad \sum_{j,l}(T_{jl} - \beta_{jl}) = 1 - \|\beta\|_1. \tag{71}$$

Plugging in yields (68). $\square$

Compared with standard GW, where $\alpha = \beta = 0$ and the bound becomes $\|\Delta\|_\infty$, the prefactor $(1 - \|\alpha\|_1)(1 - \|\beta\|_1)$—the product of the mass kept on each side after masking—reduces the perturbation sensitivity by the same fraction.

**Cancellation of covered contaminated interactions.** Let $\Omega \subseteq ([n] \times [m]) \times ([n] \times [m])$ be a set of contaminated interactions, that is, index quadruples $((i,k),(j,l))$ along which $A_{ij,kl}$ has been corrupted. We say that index sets $S_\alpha, S_\beta \subseteq [n] \times [m]$ *cover* $\Omega$ if every $((i,k),(j,l)) \in \Omega$ satisfies $(i,k) \in S_\alpha$ or $(j,l) \in S_\beta$.

**Proposition C.8** (Exact cancellation under a covering mask). *Suppose $S_\alpha, S_\beta$ cover $\Omega$ and the budgets satisfy*

$$\sum_{(i,k) \in S_\alpha} T_{ik} \le \epsilon_1, \qquad \sum_{(j,l) \in S_\beta} T_{jl} \le \epsilon_2. \tag{72}$$

*Then the saturating choice*

$$\alpha_{ik} = T_{ik}\,\mathbf{1}\{(i,k) \in S_\alpha\}, \qquad \beta_{jl} = T_{jl}\,\mathbf{1}\{(j,l) \in S_\beta\} \tag{73}$$

*is feasible—that is, $(T, \alpha, \beta) \in \mathcal{D}$—and every contaminated quadruple gives zero contribution to the SRGW objective:*

$$(T_{ik} - \alpha_{ik})(T_{jl} - \beta_{jl}) = 0 \qquad \text{for all } ((i,k),(j,l)) \in \Omega. \tag{74}$$

*Proof.* We first check $(T, \alpha, \beta) \in \mathcal{D}$. The transport plan is unchanged, so $T \in \Pi(\mu, \nu)$. From (73), $\alpha_{ik} \in \{0, T_{ik}\}$ and $\beta_{jl} \in \{0, T_{jl}\}$, so the box constraints $0 \le \alpha \le T$ and $0 \le \beta \le T$ hold entrywise. The budget constraints follow from (72):

$$\|\alpha\|_1 = \sum_{(i,k) \in S_\alpha} T_{ik} \le \epsilon_1, \qquad \|\beta\|_1 = \sum_{(j,l) \in S_\beta} T_{jl} \le \epsilon_2. \tag{75}$$

We next prove (74). By the saturating choice (73),

$$T_{ik} - \alpha_{ik} = \begin{cases} 0, & (i,k) \in S_\alpha, \\ T_{ik}, & (i,k) \notin S_\alpha, \end{cases} \tag{76}$$

and the same holds for $T_{jl} - \beta_{jl}$ with respect to $S_\beta$. Pick any $((i,k),(j,l)) \in \Omega$. Since $S_\alpha, S_\beta$ cover $\Omega$, we have $(i,k) \in S_\alpha$ or $(j,l) \in S_\beta$. In the first case $T_{ik} - \alpha_{ik} = 0$; in the second case $T_{jl} - \beta_{jl} = 0$. The product is therefore zero, which proves (74). $\qquad \square$

Proposition C.8 shows that the masks can remove an entire contaminated set $\Omega$ from the SRGW objective without turning the transport problem into an unbalanced one: $T$ stays in $\Pi(\mu, \nu)$, and the removal happens purely through the multiplicative weights $T - \alpha$ and $T - \beta$.

**Corollary C.9** (Robustness to covered corruption). *Let $\widetilde{A} = A + \Delta$ where $\Delta$ is supported on $\Omega$ (i.e., $\Delta_{ij,kl} = 0$ for $((i,k),(j,l)) \notin \Omega$). If $S_\alpha, S_\beta$ cover $\Omega$ and the budgets satisfy (72), then under the saturating mask (73),*

$$\Phi_{\widetilde{A}}(T, \alpha, \beta) = \Phi_A(T, \alpha, \beta). \tag{77}$$

*Proof.* Since $\Delta$ is zero outside $\Omega$,

$$\Phi_{\widetilde{A}} - \Phi_A = \sum_{((i,k),(j,l)) \in \Omega} \Delta_{ij,kl}(T_{ik} - \alpha_{ik})(T_{jl} - \beta_{jl}), \tag{78}$$

and (74) makes every term in the sum zero. $\qquad \square$

Corollary C.9 strengthens Proposition C.7 when the corruption sits inside a coverable set: the bound in Proposition C.7 becomes an exact equality. The two results together explain what the budgets buy: any uncovered corruption is paid for at the rate $\|\Delta\|_\infty(1 - \|\alpha\|_1)(1 - \|\beta\|_1)$ from Proposition C.7, while any covered part is removed exactly.

**Low subgraph ratios.** At very small sampling ratio $\rho$, both the transport gradient and the mask gain are driven by limited pairwise evidence. For the $\alpha$-gain

$$G_{ik}^{\alpha} = \sum_{j,l} L(C_{ij}, D_{kl})(T_{jl} - \beta_{jl}), \tag{79}$$

fix a source node $i$, its true target $a = \pi(i)$, and a wrong candidate $k \neq a$. The gain margin is

$$\begin{aligned} \Delta_i^{\alpha}(k) &:= G_{ik}^{\alpha} - G_{ia}^{\alpha} \\ &= \sum_{j,l} \left[ L(C_{ij}, D_{kl}) - L(C_{ij}, D_{al}) \right] (T_{jl} - \beta_{jl}). \end{aligned} \tag{80}$$

Let $R = T - \beta$ and $p_l = \sum_j R_{jl}$, so that $\sum_l p_l \leq 1$. Under uniform target marginals $\nu_l = 1/N$, the true target support has size $\rho N$, and $p_l \leq \nu_l = 1/N$ for every $l$, so $\sum_{l \in \mathrm{supp}(\pi)} p_l \leq \rho N \cdot (1/N) = \rho$. The clean support therefore contributes only $O(\rho)$ effective mass to the gain margin, so the signal-to-noise ratio between true and wrong candidates gets worse as $\rho \to 0$, which explains the weaker performance observed at very low subgraph ratios.

