# OpenReview forum: "Achieving Structurally Robust Gromov Wasserstein Distance via Adaptive Dual-Mask"
_ICML.cc/2026/Conference — ICML 2026 regular_

### Official Review · Reviewer_HLRW · 2026-03-11

**Soundness:** 3
**Presentation:** 2
**Significance:** 2
**Originality:** 2
**Overall Recommendation:** 4
**Confidence:** 3

**Summary:**

This paper introduces the Structurally Robust Gromov-Wasserstein (SRGW) distance to improve optimal transport on graphs corrupted by edge-induced structural noise. It utilizes an adaptive dual-mask mechanism to surgically filter out spurious edge connections while preserving valid inlier nodes. Optimized via a highly efficient Mask-Guided GW algorithm, this approach achieves state-of-the-art topological recovery and runs significantly faster than existing robust baselines.

**Compliance With Llm Reviewing Policy:**

Affirmed.

**Key Questions For Authors:**

-	Eq. 2:
    -	will T-\alpha and T-\beta become negative? If negative, will it destory the quadratic formulation of GW?
    -	Compared to substracting alpha and beta, can we formulate alpha and beta as some masking multipled on T? i.e., $T_{i,j}\alpha_{i,j}T_{k,l}\beta_{k,l}$.
-	Figure 4(c): it is shown that the performance degrades significantly when $\epsilon$ is set to be too large. Can you explain potential cause of this? For example, what would happen for extreme cases when $\epsilon$ is set to be inf? Or when $\epsilon$ is set to be 0?

**Limitations:**

Yes

**Strengths And Weaknesses:**

**Strength**

-	Novel method: By shifting from node-wise mass disposal to edge-wise interaction filtering, SRGW alleviates the error propagation problem when facing noisy data.
-	Computational Efficiency: Formulating the mask-updating step as a Continuous Knapsack Problem enables a closed-form, greedy solution.
-	Theoretical Guarantees: The paper provides a thorough mathematical proof demonstrating whole-sequence convergence to a stationary point

**Weakness**

-	Hyperparameter Sensitivity: The method relies heavily on predefined total noise budgets ($\epsilon_1, \epsilon_2$).
-	Scalability Results: empirical scalability evaluation is strongly recommended, e.g., how the run time / computational cost changes when the graph size increases.
-	Experiments: the paper mostly focus on the subgraph alignment task. I would recommend some experiments on graph-level tasks, e.g., graph classification/clustering to evaluate the quality of the metric. Besides, it is recommended to test the scalability to larger datasets to evaluate the scalability.

---

> ### Author Rebuttal · Authors · 2026-03-31
>
> ## Response to Reviewer HLRW
>
> We sincerely thank you for recognizing the novelty of our edge-wise filtering approach, the computational efficiency of our continuous knapsack formulation, and our theoretical convergence guarantees. We address your specific concerns below.
>
> **1. Eq. 2: Will $(T-\alpha)$ and $(T-\beta)$ become negative? Does it destroy the quadratic formulation?**
>
> They will strictly never become negative. Our optimization framework explicitly enforces capacity constraints on the mask variables: as defined in the feasible set $\mathcal{D}$, we have $0\le\alpha\le T$ and $0\le\beta\le T$, which our Mask-Guided GW algorithm directly maintains during the continuous knapsack update. Since the masks can never exceed the transport mass at any index, $(T-\alpha)$ and $(T-\beta)$ remain strictly non-negative, fully preserving the probabilistic validity and quadratic structure of the GW formulation.
>
> **2. Formulating masks multiplicatively (e.g., $T\odot M$) instead of subtractively:**
>
> We thank the reviewer for this suggestion. A multiplicative view is indeed possible: since $0 \le \alpha \le T$ and $0 \le \beta\le T$, one can reparameterize with retention masks $M^\alpha_{ik}=1-\alpha_{ik}/T_{ik}$ and $M^\beta_{jl}=1 - \beta_{jl}/T_{jl}$, so that $(T_{ik}- \alpha_{ik})(T_{jl}-\beta_{jl})= T_{ik}M^\alpha_{ik}\,T_{jl}M^\beta_{jl}$. The two formulations are thus largely equivalent. We adopt the subtractive parameterization because it offers a clearer interpretation of $\alpha,\beta$ as attenuated interaction mass while keeping $T$ under strict marginal constraints, and crucially, it yields simple mask-update subproblems admitting efficient greedy/proximal solutions.
>
> **3.Regarding Hyperparameter Sensitivity  $\epsilon_1,\epsilon_2$**
>
> We clarify that $\epsilon_1,\epsilon_2$ are not arbitrary regularization parameters but carry direct structural meaning: they bound the total mask mass $(\alpha,\beta)$, encoding the prior on the fraction of corrupted structure—a semantically meaningful quantity rather than an opaque tuning knob. In the subgraph setting,   $\epsilon$ is tied directly to the known sampling ratio, yielding a parameter-free default used in all experiments. When the corruption level is unknown, $\epsilon$ can be selected via a lightweight sweep around the anticipated retained-structure ratio, or by unsupervised proxy criteria measuring reconstruction consistency under the learned coupling. Figure 5(c) confirms that performance remains stable across a wide range of ε values around the true corruption level.
>
> **4. Figure 5(c) : Explanation of performance degradation at extreme $\epsilon$ values:**
>
> The parameter $\epsilon$ controls the global budget allocated for structural noise filtering via the masks α and β. Performance peaks near the true noise ratio and degrades at both extremes for the following reasons:
>
> - **When $\epsilon$ → 0:** The budget forces both masks to $\alpha =\beta=0$, reducing SRGW exactly to standard GW with no noise suppression. The model is fully exposed to structural distortions, causing accuracy to drop.
> - **When $\epsilon$ → 1:** The optimizer exploits the excessive budget to mask out most edge interactions, collapsing to a trivial near-zero transport cost that destroys geometric alignment. This explains the sharp accuracy drop on the right side of the curve.
>
> The optimal $\epsilon$ lies near the true inlier ratio, acting as a well-calibrated prior on the corruption level. In practice, ε can be set heuristically from a rough estimate of the noise ratio, which is often available, making the method practically applicable despite this sensitivity.
>
> **5. Scalability Results and Graph-Level Tasks (Weaknesses):**
>
> We thank the reviewer for the constructive feedback and have conducted additional experiments to address the key concerns.
>
> **Scalability.** Table 1 reports average runtime for subgraph matching on ModelNet across varying graph sizes. Our method  scales favorably compared to RGW, which grows substantially faster with graph size, while remaining competitive with UGW.
>
> **Table 1.** Average runtime (seconds) on ModelNet.
>
> | Method | 50       | 100      | 200      | 500   |
> | ------ | -------- | -------- | -------- | ----- |
> | Ours   | **0.39** | **0.53** | **0.78** | 0.87  |
> | RGW    | 4.66     | 18.51    | 23.38    | 64.42 |
> | UGW    | 0.58     | 0.84     | 1.07     | 1.25  |
>
> **Graph Classification.** We evaluate graph-level classification on ENZYMES using pairwise GW distances as a kernel for SVM. Our method achieves the highest accuracy , outperforming UGW , PGW , and RGW, demonstrating that our structural metric is more informative for downstream tasks under partial graph sampling.
>
> **Table 2.** Classification accuracy  on ENZYMES.
>
> | Method | Accuracy   |
> | ------ | ---------- |
> | Ours   | **0.7917** |
> | UGW    | 0.7396     |
> | PGW    | 0.4792     |
> | RGW    | 0.2188     |

---

> > ### Author Rebuttal · Reviewer_HLRW · 2026-04-04
> >
> > The authors have addressed my concerns. I will keep my ratings and lean to acceptance.

---

> > > ### Author Response · Authors · 2026-04-05
> > >
> > > Thank you for your positive feedback and your continued support for our paper. We are thrilled that our response successfully addressed your concerns. We will thoroughly integrate the discussions from this rebuttal into the final version of the manuscript to further improve the paper. Thank you once again for your constructive reviews and your time!

---

### Official Review · Reviewer_grBw · 2026-03-13

**Soundness:** 3
**Presentation:** 3
**Significance:** 3
**Originality:** 2
**Overall Recommendation:** 4
**Confidence:** 3

**Summary:**

This paper studies the robustness of the Gromov-Wasserstein (GW) distance under structural noise. The authors argue that existing robust GW methods mainly rely on node-centric mass relaxation, which can handle node outliers but fails to address edge-level distortions caused by corrupted relationships between valid nodes. To address this issue, the paper proposes a new formulation called Structurally Robust Gromov-Wasserstein (SRGW). The key idea is to introduce two auxiliary masking variables that filter unreliable interactions during the optimization of the quadratic GW objective. The resulting problem is solved using an alternating block coordinate descent algorithm (Mask-Guided GW), where the transport plan and masking variables are updated iteratively. The authors also provide a convergence analysis under the Kurdyka–Łojasiewicz framework and demonstrate empirical improvements on synthetic geometric matching and graph alignment tasks.

**Compliance With Llm Reviewing Policy:**

Affirmed.

**Final Justification:**

I sincerely thank the authors for their detailed rebuttals, which answered my questions. I will increase the score. I hope to revise the relevant rebuttals in the final manuscript.

**Key Questions For Authors:**

Referring to the aforementioned Weaknesses and Concerns.

**Limitations:**

yes

**Strengths And Weaknesses:**

Strengths:

1、 Robustness of the Gromov-Wasserstein distance under structural noise is an important topic, particularly for graph matching, geometric correspondence, and structured data alignment.

2、The paper provides an intuitive explanation of how structural noise can propagate globally in the quadratic GW objective.

Weaknesses：

1、The paper claims to shift from node-centric robustness mechanisms to an edge-centric interaction filtering perspective. However, the actual formulation suggests that the masking variables are still defined over node correspondences rather than true edge interactions. Specifically, the objective modifies the quadratic GW term as

$\sum L(C_{ij}, D_{kl}) (T_{ik}-\alpha_{ik})(T_{jl}-\beta_{jl}),$

where the auxiliary variables $\alpha_{ik}$ and $\beta_{jl}$ are indexed by node pairs $(i,k)$. As a result, the masking mechanism effectively reweights the transport variables $T_{ik}$ rather than directly modeling reliability at the edge level $(i,j,k,l)$.

2、The paper provides convergence guarantees under the Kurdyka-Lojasiewicz framework. However, there appears to be a mismatch between the algorithm used in practice and the one analyzed theoretically. In the implementation, the masking variables are updated using a greedy continuous knapsack solver, where entries are sorted according to gain values and the budget is allocated sequentially. In contrast, the theoretical analysis assumes proximal mask updates of the form

$\alpha^{k+1} =\Pi_{A(T^{k+1})}\left(\alpha^k + \frac{1}{\delta} G^{\alpha}\right),$

which corresponds to a projected proximal step with $\delta > 0$. Since the greedy solver corresponds to the limiting case $\delta \to 0$, the theoretical convergence guarantees do not directly apply to the algorithm used in the experiments.

3、The proposed masking variables appear conceptually related to existing robustness mechanisms in the optimal transport literature. Several prior works introduce slack variables, relaxed marginal constraints, or perturbation variables to absorb outliers and structural noise in GW-type objectives. From an optimization perspective, the proposed formulation resembles standard robustification strategies for quadratic assignment or robust optimal transport. The paper would benefit from a clearer discussion of how the proposed approach fundamentally differs from existing robust GW formulations.

Minor Concerns：

1、The experimental section mainly evaluates synthetic geometric matching and graph alignment tasks. Including additional benchmarks (e.g., shape correspondence or other 3D geometric matching datasets) would further strengthen the empirical evaluation.

2、The selection of the masking budgets $\epsilon_1$ and $\epsilon_2$ is not clearly discussed. It would be helpful to provide guidance on how these hyperparameters are chosen in practice.

3、 The computational complexity and runtime of the proposed method relative to existing GW solvers could be discussed more explicitly.

---

> ### Author Rebuttal · Authors · 2026-03-31
>
> ### Response to Reviewer grBw
>
> We thank the reviewer for the careful and constructive feedback, and especially for recognizing the importance of robustness under structural noise and the intuitive explanation of how local corruption propagates through the quadratic GW objective. We will revise the paper accordingly.
>
> **1.W1: Interaction-Level Masking**
> We agree that our method does not introduce an explicit 4-way reliability tensor over all $(i,j,k,l)$ interactions. However, our point is precisely that such an additional tensor is not necessary. In GW, edge-to-edge interactions are already induced by the quadratic coupling term $T_{ik}T_{jl}$. Therefore, rather than introducing a separate full mask $M_{ijkl}$, we modulate the two coupling factors directly through $(T_{ik}-\alpha_{ik})(T_{jl}-\beta_{jl})$. This yields a tractable factorized surrogate for interaction-level filtering: it still acts inside the quadratic GW term, but avoids the substantial storage and optimization overhead of a full 4-way interaction mask.
>
> **2.W2: Theory–Implementation Gap**
>
> The implementation uses a greedy continuous-knapsack solver corresponding to the limiting case $\lambda\to 0$, which falls outside Lemma 4.3 as stated. We address this directly.
>
> Under two mild conditions—(i) feasibility stability of the capped-simplex sets, and (ii) nondegeneracy of greedy solutions (Feasibility stability holds asymptotically as $T^k$ converges, and nondegeneracy is generic under continuous perturbations.)—exact knapsack optimality yields an LP descent bound:
>
> $\langle G_\alpha^k,\alpha^{k+1}-\alpha^k\rangle\geq c\|\alpha^{k+1}- \alpha^k\|_1\gtrsim\|\alpha^{k+1}-\alpha^k\|_F^2$
>
> and analogously for $\beta$. This replaces proximal strong convexity in Lemma 4.3 without requiring $\lambda>0$. The exact KKT conditions of the greedy subproblems further recover the relative-error bound of Lemma 4.4 (up to Lipschitz terms), so the KL-based convergence argument carries through after a finite burn-in.
>
> **We retained the proximalized presentation** as it handles the moving feasible sets $0\leq\alpha,\beta\leq T$ in a unified, assumption-free manner. Nevertheless, we will add a formal corollary explicitly confirming that greedy knapsack updates satisfy the same convergence guarantees, closing the theory-practice gap.
>
> **3.W3: Prior Robust GW Methods**
>
> Existing methods (UGW, PGW,srGW, RGW) relax marginal constraints to handle node-level corruptions, but under structural corruption this becomes counterproductive: high-distortion nodes should be retained in the matching, and discarding them via mass relaxation conflates structural noise with genuine absence of correspondence.
>
> Our method instead filters high-distortion interactions directly inside the quadratic GW term while retaining hard marginal constraints. When a node is globally corrupted across all its relations, the masking variables consistently suppress its interactions—achieving the same effect as marginal relaxation without a separate relaxation mechanism.
>
> **4.Benchmark Results**
>  We thank the reviewer for this suggestion. We have conducted additional experiments on **ModelNet10**, a standard 3D shape benchmark, evaluating subgraph shape matching under structural noise.
>
> **Results on ModelNet10 (Subgraph Shape Matching).**
>
> | Method    | Top-1 Acc  | Top-5 Acc  |
> | --------- | ---------- | ---------- |
> | UGW       | 0.5104     | 0.8802     |
> | PGW       | 0.4844     | 0.6979     |
> | RGW       | 0.5365     | 0.7812     |
> | SRGW      | 0.3021     | 0.6510     |
> | **Ours ** | **0.6979** | **0.9323** |
>
> Our method achieves the highest Top-1 accuracy (0.6979 vs. 0.5365 for the best baseline RGW, a **+30% relative improvement**) and the best Top-5 accuracy (0.9323).
>
> **5.Budget Parameters**
> The budgets $\epsilon_1,\epsilon_2$ bound the dual mask mass $(\alpha,\beta)$, encoding the prior on the fraction of corrupted structure. In the subgraph setting we tie $\epsilon$ to the known sampling ratio as a principled default; when this is unavailable, a lightweight sweep around the anticipated retained-structure ratio suffices. We will add this guidance to the revision.
>
> **6.Regarding complexity and runtime.**
>
> In the dense setting, the dominant cost remains the GW-type tensor contraction, computable in $O(n^2m + nm^2)$ via the squared-loss simplification. Each iteration consists of one $T $-step and two mask updates (each requiring one contraction plus $O(nm\log nm)$ sorting), giving overall complexity $O(n^2m + nm^2)$ per iteration—the same leading order as BAPG. The dual-mask mechanism thus adds only a constant-factor overhead while explicitly filtering structural noise. Compared to UGW and RGW, which require additional Sinkhorn subproblems, our method enjoys lower practical runtime; compared to lighter solvers (PGW, FW, BAPG), it trades a modest constant overhead for substantially better accuracy–robustness under structural noise.

---

> > ### Author Rebuttal · Reviewer_grBw · 2026-04-04
> >
> > I sincerely thank the authors for their rebuttal, which has addressed my concerns. I will increase the score. If the paper gets accepted, I hope to revise the relevant rebuttals in the final manuscript.

---

> > > ### Author Response · Authors · 2026-04-05
> > >
> > > Thank you very much for your positive feedback, your support, and for increasing the score. We are glad our response resolved your concerns. We assure you that all relevant revisions and clarifications discussed in the rebuttal will be carefully integrated into the final version of the manuscript. Thank you again for your time and valuable suggestions.

---

### Official Review · Reviewer_U34a · 2026-03-18

**Soundness:** 3
**Presentation:** 2
**Significance:** 2
**Originality:** 3
**Overall Recommendation:** 5
**Confidence:** 5

**Summary:**

The authors introduce a novel method for robust graph matching based on the Gromov-Wasserstein (GW) distance. Authors argue that most methods focus on node-level noise (outliers) via marginal relaxation but fail to handle edge-level structural noise between valid nodes. Consequently, they propose Structurally Robust Gromov-Wasserstein (SRGW), which introduces a dual-mask mechanism that adaptively filters structural distortions during optimization while preserving balanced transport constraints. The method jointly learns a transport plan between structures and masking variables over an OT plan that identifies and suppresses noisy structural interactions, using a tailored block-coordinate algorithm. The authors provide a theoretical convergence guarantee of a soft variant of their algorithm. Finally, they show that SRGW outperforms SOTA GW-based methods for subgraph matching and graph matching with structural noise on synthetic tasks derived from 4 real-world datasets.

**Compliance With Llm Reviewing Policy:**

Affirmed.

**Final Justification:**

Dear authors, thank you for these detailed explanations. I believe that the results in your last reply are important to integrate in the main paper. In particular I encourage authors for the question 3 to add some details in the proof to ease readibility, clearly expressing that : 1) you proved here that knowing the contamination edges, *if mass is assigned to them via T*, then it is optimal to cancel it via the mask variables as much as possible, knowing the constraints are satured (as implied by the closed form solutions); 2) add explanations regarding why it is "enough" to prove this statement, covering cases where mass is not assigned to these contaminated edges then T would be suboptimal.

It is also important to integrate new results covered in your first answer including:
- correct all presentation issues as promised by authors.
- modifications of assumptions for convergence analysis as a complementary result which is more aligned with your empirical study.
- sensitivity analysis to the type of pairwise matrices considered.
- sensitivity analysis to the hyperparameters in comparison to the other most competitive methods.
- honest analysis on marginal deviations, turning down some of the claims in the main paper.

Overall I consider authors' rebuttal very compelling. Therefore, I will increase my score from 3 (Weak reject) to 5 (accept).

**Key Questions For Authors:**

I strongly encourage authors to discuss the weaknesses identified above. Follows a few questions to clarify specific points :

  -	*[addressed]* Could you provide intuitions onto why SRGW seems to work relatively less well with low subgraph selection ratios?
  -	*[addressed]* Don’t you think that it would be feasible to extend the proof, e.g., of Lemma 4.3 (and the other results too) by considering exact solvers for the dual-mask variables while leveraging the closed-form solutions, which should allow you to bound the differences between objectives across iterations? The proximalized variants allow us to leverage strong convexity, which seems to be a bit overkill for those simple problems.

**Limitations:**

Yes

**Strengths And Weaknesses:**

**NB**: comments written in italic were made after authors' rebuttal.

**Strengths**

  -	Introduce a novel OT problem relevant for handling noisy graph matching
  -	Introduce a novel BCD solver alternating between a BAPG to solve the subproblems w.r.t the transport plan and leverage closed-form solvers for the 2 independent linear problems w.r.t the dual-mask variables.
  -	Study the convergence of a related optimization problem relying Kurdyka–Łojasiewicz property, modifying the closed-form solvers with proximal solvers.  The proofs seemed correct to me.
  -	show that SRGW outperforms SOTA GW-based methods for subgraph matching (over various subgraph size ratios) and matching on structural noise on synthetic tasks derived from 4 real-world datasets,


**Weaknesses**

**W1. Presentation clarity**. *[addressed - authors promised to make these modifications in their revised version]* There are many typos and points that could benefit from being clarified in the main paper to ease the reading of the paper.

  -	a) In Figure 1 : It’ll be much clearer to illustrate the matching between a graph and itself, vs a graph and a corrupted version. Or simply explain with a sentence that the former results in a node-wise structural stress of 0 vs not. Illustrating only the second matching if you don’t have space.
  -	b) The presentation of the UGW and PGW could be made clearer under the unified unbalanced GW framework discussed in [A], where the difference between “UGW” and “PGW” essentially lies in the choice of marginal penalizations.
  -	c) The set of constraints on $\alpha$ and $\beta$ in Equation 2 is inaccurate and does not coincide with the ones discussed in the following equations.
  -	d) No reference provided for several results relating to Continuous Knapsack problem e.g solutions of eq.5 or eq.8
  -	e) The complexity of their algorithm should be discussed by the authors.
  -	f) $\psi$ not defined in Theorem 1, RAPG not defined ?
  -	g) Some Tipos: L26 (right) on -> in / L60 (left) weird repetition with Figure 1. / L202 (right) bilinear -> linear ? .. And many others, so the authors should re-read their paper more carefully and correct those.

**W2. Soundness of the theoretical results**

  -	a) *[addressed]* The authors provided theoretical convergence results for a variant of their algorithm, which includes proximalized  Knapsack solvers instead of exact ones. Therefore, they do not have convergence proofs for the algorithm they empirically used, nor benchmarked its variant with proximalized solvers. The authors should further discuss why these two last statements are not covered in the paper. For now, it clearly calls into question the relevance of the authors' theoretical analysis.

  -	b) *[addressed]* The authors do not discuss how strong the assumptions they made for Theorem 1 are. In particular, Assumption C3 seems clearly not straightforward to check.

  -	c) *[addressed]* There are no theoretical results on the robustness of their proposed approach, contrary to the UGW problem, which has been shown to be robust to “node-level” noise in [C]. It would have been of the utmost relevance to study theoretically the robustness of their solvers to such noise as well as structural noise.

**W3. Comprehensiveness of the benchmark:** Some flaws that should be addressed by authors:

  -	a) *[addressed]* The robust matching problem proposed by [B] also introduces a deviation over the pairwise matrices instead of on the marginals, hence it is conceptually very similar to SRGW and should therefore be discussed and benchmarked in the main paper.
  -	b) *[addressed]* All experiments made by the authors focus on pairwise Euclidean distance matrices. These are known to lead to a specific subset of GW problems, which are concave. It would be relevant to expand the benchmark by considering more general structures, such as graph adjacency matrices commonly used in ML applications.
  -	c) *[addressed]* It seems that SRGW is highly sensitive to the parameter epsilon, which correlates with the quantity of added noise. Authors should discuss whether this parameter could be selected in an unsupervised way, and also study whether the most competitive methods are as sensitive to their parameters as SRGW. Moreover, the authors didn’t discuss the sensitivity of their method to the lambda hyperparameter in BAPG.
  - d) *[addressed]* Authors strongly argue that learning deviations in terms of pairwise structures is more efficient for the studied problems than learning deviations in terms of node mass. However the used BAPG solver is known as the GW solver that sacrifies the most feasibility of the constraint sets, i.e that leads to the largest deviations from the marginal constraints. Authors should discuss this point, and quantify the marginal deviations of their solvers.

[A] Bai, Y., Martin, R. D., Kothapalli, A., Du, H., Liu, X., & Kolouri, S. (2024). Partial Gromov-Wasserstein Metric. arXiv preprint arXiv:2402.03664.

[B] Liu, W., Xie, J., Zhang, C., Yamada, M., Zheng, N., & Qian, H. (2022, September). Robust graph dictionary learning. In The Eleventh International Conference on Learning Representations.


[C] Tran, Q. H., Janati, H., Courty, N., Flamary, R., Redko, I., Demetci, P., & Singh, R. (2023, June). Unbalanced co-optimal transport. In Proceedings of the AAAI Conference on Artificial Intelligence (Vol. 37, No. 8, pp. 10006-10016).

---

> ### Author Rebuttal · Authors · 2026-03-31
>
> ### Response to Reviewer U34a
>
> We thank the reviewer for the constructive feedback. Due to space limits, we address the main theoretical and empirical concerns here; all presentation issues will be incorporated in the revision; the full table of results (W3) is available at: https://anonymous.4open.science/r/sadadadaefdfwesadafaqw/exp.md.
>
> **1.Theory–Practice Gap (W2a)**
>
> An exact-greedy convergence result is indeed feasible under two mild assumptions: (i) feasibility stability of the capped-simplex sets as $T^k $ stabilizes, and (ii) nondegeneracy of greedy solutions (generic under continuous perturbations). Under these, exact knapsack optimality yields an LP descent bound $\langle G_\alpha^k,\alpha^{k+1}-\alpha^k\rangle\geq c\|\alpha^{k+1}-\alpha^k\|_1\gtrsim\|\alpha^{k+1}-\alpha^k\|_F^2$, replacing proximal strong convexity in Lemma 4.3, and the exact KKT conditions recover the relative-error bound of Lemma 4.4 (up to Lipschitz terms), so the KL argument still applies.
>
> We adopted the proximalized presentation as it handles the moving feasible sets $0\le\alpha,\beta\le T $ in a unified manner without auxiliary assumptions, at the cost of negligible regularization. We will include an appendix sketch in the revision.
>
> **2.Marginal Feasibility (W3d)**
>
> **Marginal Feasibility.** On ENZYMES, ours method yields RowDev/ColDev of 1.99e-01/4.19e-08, matching BAPG (1.84e-01/5.43e-08) on both marginals—confirming no additional feasibility issue from the mask-guided design. Unbalanced alternatives incur far larger deviations: rgw (2.34e-01/6.18e-01), ugw (1.38e-01/1.52e-01). The residual row deviation is a necessary cost of maintaining alignment accuracy under unequal graph sizes. These results will be included in the revision.
>
> **3.Expanded Benchmarks (W3a,b)**
>
> **Adjacency Matrix Experiments (W3a).** We clarify that our experiments use pairwise Euclidean distances—not squared Euclidean distance matrices, so the concave GW setting does not directly apply. That said, we re-ran Subgraph Alignment with raw adjacency and shortest-path matrices as inputs: ours method consistently outperforms all baselines. On ENZYMES with shortest-path matrices, our method achieves $56.58%$ vs.$45.54%$ for the best baseline, confirming generalization beyond the Euclidean setting. Full results will be included in the revision.
>
> **Comparison with Baseline [B] (W3b).**  We have added RGDL as a baseline. Accuracies: Ours 0.524, rgw 0.416, RGDL 0.109 on ENZYMES. Although both methods perturb pairwise matrices, RGDL targets graph dictionary learning—its minimax  formulation suppresses local topological signatures needed for node matching. RGDL's lowest accuracy confirms this task mismatch. This discussion will be added to the revision.
>
> **4. Sensitivity (W3c&q1)**
>
> - **Sensitivity to  $\lambda$  and $\epsilon$ **
>
>   **Parameter $\lambda$ .** On ENZYMES, $\lambda\in\{0.005,0.01,0.05,0.1\}$ yields accuracies of $0.535,0.566,0.523,0.438$. For reference, RGW has seven  hyperparameters vs. our three, and shows comparable sensitivity to its key parameter $\eta $  (accuracies $0.411,0.389,0.398,0.354$ over $\eta\in\{0.005,0.01,0.05,0.1\}$, making our method substantially easier to tune in practice.
>
>   **Parameter $\epsilon$.** The budget $\epsilon$ bounds the dual mask mass $(\alpha,\beta)$, encoding the prior on corrupted-structure fraction. In the subgraph setting we tie $\epsilon$ to the known sampling ratio; otherwise a lightweight sweep around the anticipated retained-structure ratio suffices. This guidance will be added to the revision.
>
> * **Low Subgraph Ratios.** Degradation at $\rho=0.1$ stems from a quadratic evidence collapse: pairwise evidence scales as $O(\rho^2n^2)$ while correspondence ambiguity remains $O(n^2)$, leaving the dual masks insufficient signal to identify corrupted interactions.
>
> **5.Assumptions & Robustness (W2b,c)**
>
> **Assumption C3.** Assumption C3  is standard in KL-based BCD frameworks (e.g., Attouch et al., 2013). In practice, we enforce this by stopping the inner BAPG solver when its stationarity residual falls below a shrinking threshold. Appendix B justifies this criterion via the Luo–Tseng error bound.
>
> **Theoretical Robustness .** We derive the following objective-level robustness bound: let $A_{ij,kl}=L(C_{ij},D_{kl})$  be the clean interaction tensor and $\tilde{A}=A+\Delta$ its perturbed counterpart. For any feasible $(T,\alpha,\beta)$,
>
> $\left|\Phi_{\tilde{A}}(T,\alpha,\beta)-\Phi_A(T,\alpha,\beta)\right|\leq\|\Delta\|_\infty\cdot(1-\|\alpha\|_1)(1-\|\beta\|_1)$
>
> This has two implications: (1) the dual-mask mechanism contracts perturbation sensitivity by$(1-\|\alpha\|_1)(1-\|\beta\|_1)\leq 1$, strictly improving over standard GW; (2) when corrupted mass lies within budgets $(\epsilon_1,\epsilon_2)$, the masks exactly cancel all contaminated interactions without relaxing marginal constraints on $T$, theoretically explaining the noise resilience in Sec. 5.2.2. This result will be included in the revision.

---

> > ### Author Rebuttal · Reviewer_U34a · 2026-04-06
> >
> > Dear authors, thank you for your detailed answers. Many of my concerns have been addressed but could you please clarify the following points so i can conclude my review:
> >
> > Could you clarify the methodology that you applied to evaluate RGDL ? In particular the direction of the matching seems to be of drastic importance.
> > Could you further detail your analysis (with calculus) to explain performance with low subgraph ratios ?
> > Could you complete your md file with the proofs for your theoreitcal robustness analysis ? In particular I don't see how to get your conclusions on the second result with $\epsilon$.

---

> > > ### Author Response · Authors · 2026-04-08
> > >
> > > Thank you for acknowledging that our previous responses addressed many of your concerns. Below, we clarify the final points.
> > >
> > > **Question 1: Regarding the evaluation methodology and the direction of matching.**
> > >
> > > For RGDL matching, we use the subgraph as the source graph $G_s$ and the complete graph as the target graph $G_t$. This choice is consistent with "subgraph-to-graph matching": RGWD is asymmetric and its robustness perturbation is applied to the second argument, so the perturbation acts on the $G_t$ side.
> > >
> > > **Question 2: Regarding the detailed analysis for the performance with low subgraph ratios.**
> > >
> > > In our previous response, we explained the low-$\rho$ degradation intuitively as a quadratic evidence collapse. We now explain this intuition more explicitly.
> > > The key point is that both update steps in our method are driven by pairwise cost term. In particular, the dual-mask gain is
> > > $$
> > > G^\alpha_{ik}
> > > =\sum_{j,l}L(C_{ij},D_{kl})(T_{jl}-\beta_{jl}),
> > > \tag{1}
> > > $$
> > > while the transport gradient has the same form
> > > $$
> > > (\nabla_T L_{\mathrm{GW}})\_{ik}=\sum_{j,l}L(C_{ij},D_{kl})(W_{jl}-\beta_{jl}).\tag{2}
> > > $$
> > >
> > > We analyze the dual-mask gain as an example. Fixing a source node $i$, let $a=\pi(i)$ be its true target, and consider a wrong candidate $k\neq a$. We define the gain margin to characterize the ability to distinguish the true candidate from a wrong candidate.
> > > $$
> > > \Delta_i^\alpha(k):=G^\alpha\_{ik}-G^\alpha_{ia}.
> > > $$
> > > By subtracting the two terms in (1), we obtain
> > > $$
> > > \Delta_i^\alpha(k)=\sum_{j,l}\big(L(C\_{ij},D_{kl})-L(C_{ij},D_{al})\big)(T_{jl}-\beta_{jl}).
> > > $$
> > > If we write
> > > $$
> > > R:=T-\beta,\qquad p_l:=\sum\_j R_{jl},
> > > $$
> > > then the margin can be grouped column-wise as
> > > $$
> > > \Delta_i^\alpha(k)
> > > =\sum\_{l=1}^N p_l\,\eta_{i,l}^{(k)},
> > > \tag{3}
> > > $$
> > > where
> > > $$
> > > \eta_{i,l}^{(k)}:=\frac{1}{p_l}\sum\_j\big(L(C_{ij},D_{kl})-L(C_{ij},D_{al})\big)R_{jl}\qquad (p_l>0).
> > > $$
> > > Now let $\Omega:=\pi([n])$ be the true support in the target graph, with $|\Omega|=n=\rho N$. Under uniform target marginals and $0\le\beta\le T$, we have
> > > $$
> > > \sum\_{l\in\Omega}p_l\le\sum_{l\in\Omega}\nu_l=\frac{|\Omega|}{N}=\rho.\tag{4}
> > > $$
> > > Denoting the per-element contribution upper bound by $\mathbb{E}[\eta_{i,l}^{(k)}\mid l\in\Omega]\le A_\alpha$, then
> > > $$
> > > \mathbb{E}\!\left[\sum_{l\in\Omega}p_l\,\eta_{i,l}^{(k)}\right]\le A_\alpha\rho.
> > > $$
> > > So the true support can contribute at most $O(\rho)$ mass to either update  regardless of the masking budget. Therefore, when $\rho$ is very small, the true candidate has only a weak advantage over wrong candidates in both update steps,  which makes wrong candidates harder to be distinguished from the true one.
> > >
> > > **Question 3: Regarding the proofs for theoretical robustness.**
> > >
> > > We apologize for the insufficient clarity in our previous response due to the words limit. Now, we provide a complete proof below.
> > >
> > > Our objective is:
> > > $$
> > > \Phi(T,\alpha,\beta)=\sum_{i,j,k,l}L(C_{ij},D_{kl})(T_{ik}-\alpha_{ik})(T_{jl}-\beta_{jl})
> > > $$
> > > with $T\in\Pi(\mu,\nu)$, $0\le\alpha,\beta\le T$, $\|\alpha\|\_1\le\epsilon_1$, and $\|\beta\|\_1\le\epsilon_2$. Let $\widetilde{A}=A+\Delta$, $\widehat{T}:=T-\alpha$, and $\widehat{W}:=T-\beta$. Since $\|\widehat{T}\|\_1=1-\|\alpha\|\_1$ and $\|\widehat{W}\|\_1=1-\|\beta\|\_1$, we have:
> > > $$
> > > |\Phi_{\tilde{A}}(T,\alpha,\beta)-\Phi_{A}(T,\alpha,\beta)|=\sum\_{i,j,k,l} \Delta\_{ij,kl}\, \hat{T}\_{ik} \hat{W}\_{jl}\le\|\Delta\|_{\infty}(1-\|\alpha\|\_1)(1-\|\beta\|\_1)
> > > $$
> > > Compared with standard GW (where $\alpha=\beta=0$), the sensitivity to perturbations is reduced by the multiplicative factor $(1-\|\alpha\|_1)(1-\|\beta\|_1)$. This is **the first implication** mentioned in our previous response for theoretical robustness.
> > >
> > > We now turn to the second part. Let $\Omega$ be the set of contaminated interactions. Suppose the index sets $S_\alpha, S_\beta\subseteq [n]\times [m]$ cover $\Omega$, i.e., every $\bigl((i,k),(j,l)\bigr)\in\Omega$ satisfies $(i,k)\in S_\alpha$ or $(j,l)\in S_\beta.$ Define $\alpha^\star\_{ik} := T\_{ik}\,\mathbf{1}\_{(i,k)\in S\_\alpha},\beta^\star\_{jl} := T\_{jl}\,\mathbf{1}\_{(j,l)\in S\_\beta},$ which are feasible by the budget conditions.
> > >
> > > For any $\bigl((i,k),(j,l)\bigr)\in\Omega,$ the covering property guarantees $(i,k)\in S_\alpha$ or $(j,l)\in S_\beta$, yielding $T_{ik} -\alpha^\star_{ik} = 0$ or $T_{jl} -\beta^\star_{jl} = 0$. Hence
> > > $$
> > > (T\_{ik} - \alpha^\star_{ik})(T_{jl} - \beta^\star_{jl}) = 0\qquad\forall\, \bigl((i,k),(j,l)\bigr) \in \Omega
> > > $$
> > >
> > > and summing over $\Omega$,
> > >
> > > $$
> > > \sum\_{\bigl((i,k),(j,l)\bigr)\in\Omega}\mathcal{A}\_{ij,kl}(T\_{ik} - \alpha^\star\_{ik})(T\_{jl} - \beta^\star\_{jl}) = 0.
> > > $$
> > >
> > > Thus, the contaminated interaction contributes exactly zero to the SRGW objective. So we say that the contamination is canceled. Meanwhile, $T$ itself still belongs to $\Pi(\mu,\nu)$, so the balanced transport constraints are preserved. This is **the second implication** mentioned in our previous response for theoretical robustness.

---

### Decision · Program_Chairs · 2026-04-30

**Decision:**

Accept (regular)

**Comment:**

The paper proposed a new method to compute Gromov Wasserstein distance that is
more robust to structural noise, that is noise on the edges of the graphs. This
is novel because most existing method rely on re-weighting the nodes (partial OT,
unbalanced) and not the
edges. The method is based on a dual mask that is learned to compensate for the
OT plans on specific edge correspondences in the GW loss. The method is shown to be more robust to structural
noise on synthetic and real data.

The reviewers found the paper interesting but had some concerns about the
clarity, the theoretical analysis and the experiments that were
limited. The authors did a detailed response, including new experiments (in
graph classification shape matching), computational time and
clarifications that were appreciated by all reviewers. They all stated that their
concerns were fully resolved. I agree that the paper is interesting and that the
method is novel so I recommend an acceptance but I expect the authors to include
in the final version all the clarifications and new experiments that they did in
the response.